# Metabolite import by SLC33A1 is required for ATF6 activation during endoplasmic reticulum stress

Ginto George[1] , Heather P Harding[1] , Richard Kay[2], David Ron[1] , Adriana Ordoñez[1,3]

The transcription factor ATF6$\alpha$ has a central role in adapting mammalian cells to ER stress via the unfolded protein response (UPR), prompting efforts to identify ATF6$\alpha$ modulators. Here, an unbiased genome-wide CRISPR-Cas9 screen performed in Chinese Hamster Ovary cells revealed that proteolytic processing of the ATF6$\alpha$ precursor to its active form was impaired in cells lacking the ER-resident solute carrier SLC33A1, a transporter previously implicated in acetyl-CoA import, sialylation, and N$\varepsilon$-lysine protein acetylation. Cells lacking SLC33A1 constitutively trafficked the ATF6$\alpha$ to the Golgi but exhibited impaired Golgi processing and activating proteolysis. IRE1$\alpha$ signalling was derepressed by SLC33A1 deficiency consistent with selective loss of ATF6$\alpha$-mediated negative feedback in the UPR. *Slc33a1-*deleted cells accumulated unmodified sialylated N-glycans, precursors to acetylated glycans, likely reflecting impaired glycan processing. Deletion of ER-localised acetyltransferases NAT8 and NAT8B, which catalyse protein N$\varepsilon$-lysine acetylation in the secretory pathway, did not replicate the ATF6$\alpha$ processing defects observed in *Slc33a1-*deficient cells. Together, our findings highlight a role of SLC33A1-mediated metabolite transport in the post-ER ATF6$\alpha$ maturation, linking small-molecule metabolism to branch-specific signalling in the UPR.

## Introduction

The ER is a central hub for protein folding and the maintenance of cellular homeostasis (1). When misfolded proteins accumulate in the ER, a cellular stress response known as the unfolded protein response (UPR) is triggered to restore protein folding capacity. The UPR is regulated by three known transmembrane sensors: inositol-requiring enzyme 1 (IRE1), activating transcription factor 6 (ATF6), and Protein kinase R-like ER kinase (PERK) (2, 3). In vertebrates, ATF6 plays a pivotal role in restoring ER homeostasis by promoting the expression of molecular chaperones and facilitating ER-associated degradation to eliminate misfolded proteins (4).

Vertebrates express two isoforms, ATF6$\alpha$ and ATF6$\beta$. ATF6$\alpha$ has a predominant role in regulating UPR target genes in mammalian cells and is the primary focus of this study. Dysregulation of ATF6$\alpha$ signalling has been implicated in the pathophysiology of neurodegenerative disorders such as Alzheimer's and Parkinson's diseases, metabolic diseases, and cancers (5). Given its critical function in cellular stress responses, extensive research has been directed towards identifying modulators of ATF6$\alpha$ signalling.

The inactive precursor of ATF6$\alpha$ is an ER-resident type II transmembrane protein. In response to ER stress, ATF6$\alpha$ translocates from the ER to the Golgi apparatus (6). There, it undergoes sequential cleavage by the luminal site-1 protease (S1P) and the intramembrane site-2 protease (S2P), releasing a cytosolic fragment (ATF6$\alpha$-N), which further translocates to the nucleus serving as a transcription factor that activates UPR target genes (7). N-linked glycosylation of ATF6$\alpha$ in the ER has been implicated in its trafficking and processing (8), suggesting a link between ATF6$\alpha$ activation and the broader network of cellular post-translational modifications involved in regulating its function.

In a recent genome-wide CHO cell-based CRISPR-Cas9 screen for modulators of ATF6$\alpha$ signalling, we identified calreticulin, an ER-resident lectin chaperone, as a selective repressor of ATF6$\alpha$ (9). By interfering with the conversion of the inactive precursor to the cleaved active form, calreticulin maintains ATF6$\alpha$ in an inactive state under physiological conditions. Loss of calreticulin function leads to increased ATF6$\alpha$ activity, suggesting that calreticulin regulates ATF6$\alpha$ and limits UPR activity under homeostatic conditions. This finding emphasises the importance of endogenous repressors of ATF6$\alpha$ in tuning the cellular response to ER stress.

The same CRISPR-Cas9 screen also identified genes whose deletion interfered with ATF6$\alpha$ activation. Although some hits, such as genes encoding the proteases known to process the inactive ATF6$\alpha$ precursor into its active form, were anticipated, the discovery that deletion of *Slc33a1* (Solute Carrier Family 33 Member A1), a gene encoding an ER-localised solute transporter, was unexpected highlighting the need for further investigation.

SLC33A1 was first identified in 1997 by Kanamori et al 10. In that study, introducing SLC33A1 into COS-1 cells enabled the formation of O-acetylated gangliosides, suggesting its involvement in the

[1]Cambridge Institute for Medical Research (CIMR), University of Cambridge, Biomedical Campus, The Keith Peters Building, Cambridge, UK    [2]Institute of Metabolic Science | Metabolic Research Laboratories, Addenbrooke's Hospital, Cambridge, UK    [3]Universidad Católica de Murcia (UCAM), HiTech, Guadalupe, Spain

Correspondence: gg505@cam.ac.uk; aordonez7@ucam.edu

acetylation of sialic acid in the carbohydrate structure of gangliosides. This effect was attributed to the limited capacity of parental COS-1 cells to import acetyl donors into their secretory pathway, alongside the ability of SLC33A1 to import acetyl-CoA (a bulky, charged molecule that cannot freely cross membranes) into the ER lumen. Notably, SLC33A1 displayed characteristics consistent with membrane transporters, such as multiple membrane-spanning domains suggesting a potential role as an ER membrane acetyl-CoA transporter. This proposed function is further supported by a recently published cryo-electron microscopy structure of the human SLC33A1 in complex with acetyl-CoA (11).

To date, SLC33A1, also known as AT-1, remains the sole ER-localised transporter identified within cells that facilitates the transport of acetyl-CoA into the ER (10), a process recognised as essential for ER protein $N^\varepsilon$-lysine acetylation, a well-established reversible post-translational modification of nuclear, cytoplasmic, and mitochondrial proteins (12). In the secretory pathway, $N^\varepsilon$-lysine acetylation may influence both the processing of N-linked glycans in the Golgi and the stability of ER-localised proteins (13, 14, 15). In addition, SLC33A1 has been implicated in providing acetyl-CoA for the acetylation of terminal sialic acid residues during the biogenesis of gangliosides (10, 11). Therefore, similar to other post-translational modifications, proper acetylation of secretory proteins is a critical step in assisting protein folding and maturation (16).

Given its localisation and diverse functional roles, SLC33A1 is well positioned to impact ER and Golgi environments, compartments that are directly involved in ATF6α processing and activation. These roles, along with the broader connections between acetyl-CoA metabolism, ER function, and physiological processes such as ageing and neurodegeneration (reviewed in references 17, 18, 19), provided a rationale to explore how SLC33A1 activity might influence ATF6α signalling pathways.

# Results

## CRISPR–Cas9 screening identifies SLC33A1 as novel regulator of ATF6α

To identify genes whose inactivation compromises ATF6α's stress responsiveness, we used an established CHO-K1 dual UPR reporter cell line harbouring XBP1s::mCherry and BiP::GFP reporters, enabling parallel monitoring of IRE1 and ATF6α activity, respectively (9, 20) (Figs 1A and S1A). The XBP1::mCherry reporter was generated using a PCAX-F-XBP1ΔDBD-mCherry construct, in which IRE1-dependent splicing of an XBP1 transcript lacking the DNA-binding domain generates a fluorescent mCherry fusion protein. The BiP::GFP reporter consisted of the *Cricetulus griseus (cg)* BiP promoter fused to GFP and contains a single ERSE-I element motif (CCAAT-N₉-CCACG), the ER stress response element most specifically and robustly activated by ATF6α (21). Both reporters exhibited low activity in unstressed parental cells and high activity in cells exposed to agents that compromise ER proteostasis (Fig S1B) (9). CRISPR/Cas9-mediated disruption of endogenous *cgATF6α* in the dual-reporter cells resulted in a modest but reproducible

reduction in BiP::GFP fluorescence relative to parental cells, even under basal conditions indicating that the ATF6α reporter (BiP::GFP signal) exhibits partial basal activity (Fig S1B).

In a genome-wide CRISPR-Cas9 screen, whose details have been previously published (9), single-guide RNAs (sgRNA) targeting *Slc33a1* were markedly enriched in ER-stressed cells with preserved XBP1::mCherry (IRE1 pathway) activity but diminished BiP::GFP (ATF6α pathway) activity (Fig 1A and B). Notably, five out of six sgRNAs targeting *Slc33a1* in the genome-wide library were progressively enriched through the two rounds of phenotypic selection for stressed cells displaying bright XBP1::mCherry and dull BiP::GFP expression (Fig 1B). The only sgRNA that was not enriched targeted the 3' end of the gene, likely failing to inactivate it (Fig S2A). These features nominated *Slc33a1* as a candidate gene whose inactivation selectively compromises ATF6α signalling.

To confirm the genotype–phenotype relationship suggested by the screen, we targeted the *Slc33a1* locus in CHO-K1 IRE1/ATF6α dual UPR reporter cell line using two distinct CRISPR-Cas9 sgRNAs (Fig S2A). Cell pools were selected based on the expression of a FACS-compatible marker co-expressed with Cas9 and the sgRNA. *Slc33a1*-targeted pooled cells exhibited reduced responsiveness of BiP::GFP to ER stress upon tunicamycin (Tm) treatment (Fig 1C), consistent with the findings of the screen. These results support the role of *Slc33a1* as a positive modulator of ATF6α-mediated ER stress signalling.

## SLC33A1 depletion has opposing effects on IRE1 and ATF6α signalling

To further investigate the role of SLC33A1 in ATF6α activation, we isolated single clones targeting *Slc33a1* from the pools above and evaluated the genetic lesions by genomic sequencing (Fig S2B). A derivative clone, *Slc33a1Δ*clnA (Allele1: *L42Wfs*72*; Allele 2: *L42Gfs*59*), was selected for subsequent studies based on the expected severe disruption of the resulting multipass transmembrane protein, because of the loss of nearly its entire predicted transmembrane helix region (Fig S2C). In addition, cell proliferation analysis using the IncuCyte system showed that deletion of SLC33A1 (in all three backgrounds tested) did not alter the proliferative capacity of the cells compared with parental controls (Fig S2D), supporting the suitability of the mutant cells for subsequent functional assays.

As observed in the pool, under basal conditions, the clonal deletion of *Slc33a*1 did not significantly impact the ATF6α reporter levels (BiP::GFP) when compared to the parental IRE1/ATF6α dual UPR reporter cells (WT) (Fig 2A). However, depletion of SLC33A1 resulted in significant basal derepression of the IRE1 pathway reporter (Fig 2A). This was not limited to the reporter: Under basal conditions, the clonal *Slc33a1Δ*clnA cells had more spliced endogenous XBP1 mRNA (XBP1s) than parental cells (Fig 2B). Furthermore, disruption of *Slc33a1* in the clonal *Slc33a1Δ*clnA cells selectively compromised responsiveness of their BiP::GFP reporter to ER stress induced by tunicamycin (Tm) and 2-deoxy-D-glucose (2DG), while maintaining responsiveness of their XBP1s::mCherry reporter (Figs 2A and S3A), mirroring the effect observed in the pool of *Slc33a1*-targeted cells. Similar findings were observed upon induction of ER stress with other agents:

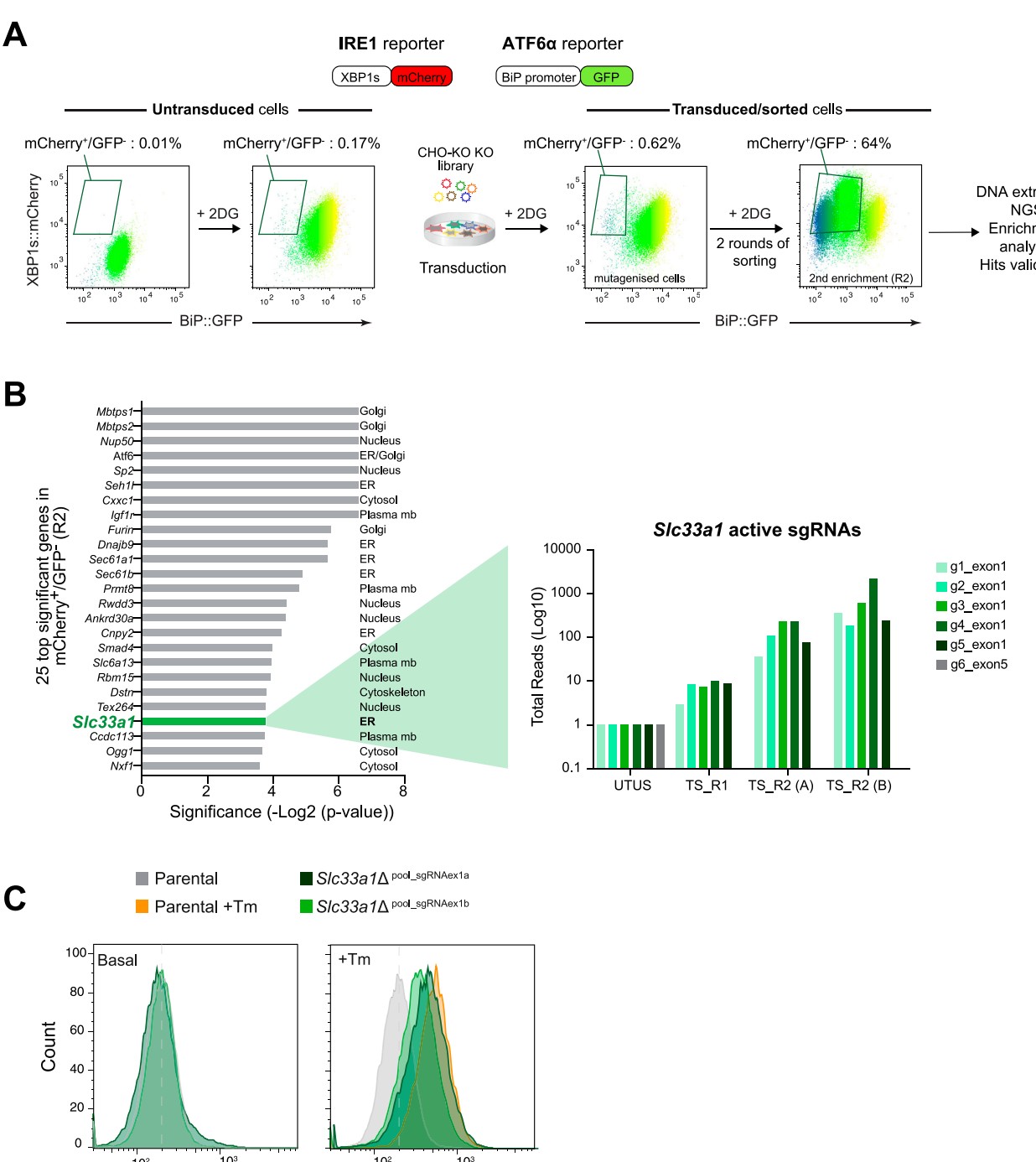

Figure 1. CRISPR-Cas9 screening identifies SLC33A1 as a positive regulator of ATF6α.

**(A)** Schematic overview of a screen for genes whose inactivation by CRISPR-Cas9 selectively compromises the ATF6α branch of the UPR. The screen was performed in dual-reporter CHO-K1 cell lines stably expressing XBP1s::mCherry (to monitor IRE1 activity) and BiP::GFP (to monitor ATF6α activity) reporters. Two-dimensional contour plots depict the reporter levels. After transduction with a genome-wide CRISPR knockout (KO) sgRNA library and treatment with 2-deoxy-D-glucose (2DG) to induce ER stress, XBP1s::mCherry-bright; BiP::GFP-dull cells recovered by FACS from the left upper quadrant of the contour plot were expanded through two rounds of enrichment (R2, sorting). Genomic DNA was extracted for next-generation sequencing (NGS), gene ontology enrichment analysis, and validation of candidate genes. **(B)** Left: significance (−log₂ [*P*-value]) provided for the top 25 genes identified by the criteria noted in (A). The *Slc33a1* gene is highlighted in green. Right: the total read count enrichment of the six sgRNAs targeting *Slc33a1* through the selection process (UTUS, untreated and unsorted; TS, treated and sorted; R1, Round 1 of enrichment; R2, Round 2 of enrichment; (A, B): pools of cells; g1–g6, sgRNAs). **(C)** Histogram of BiP::GFP reporter intensity in parental cells and two derivative *Slc33a1*-deleted (*Slc33a1Δ*) polyclonal pools under basal conditions or after ER stress induced with 2.5 µg/ml tunicamycin (Tm) for 6 h. A representative dataset from one of one single experiment is shown.

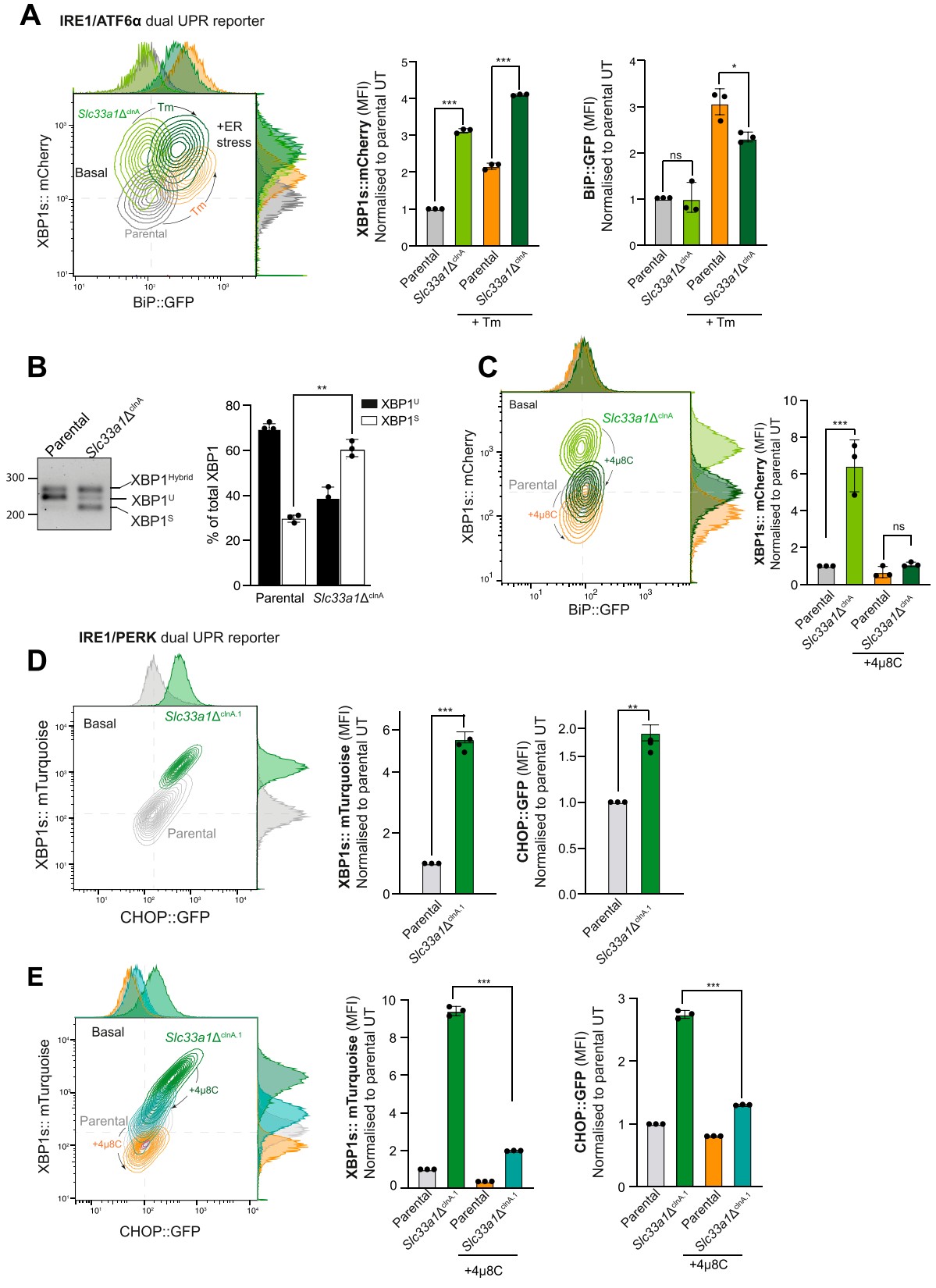

dithiothreitol (DTT) and thapsigargin (Tg) (Fig S3B and C). These findings were robust: observed in an independent, second *Slc33a1*-deleted clone (Fig S3D; *Slc33a1*Δ[clnH]) and extended to the endogenous BiP gene (Fig S3E).

To confirm that the elevated XBP1s::mCherry reporter activity observed basally in the mutant cells was driven by IRE1's RNase activity, parental and *Slc33a1*-deleted cells were exposed to 4µ8C, an IRE1 RNase inhibitor (23). Treatment with 4µ8C effectively restored XBP1s::mCherry basal levels in *Slc33a1*-deleted cells, confirming that hyperactivation of the IRE1 pathway observed in SLC33A1-targeted cells proceeded along the conventional pathway (Fig 2C).

To further explore the role of SLC33A1 in ER homeostasis, we investigated whether depletion of SLC33A1 would affect the PERK branch of the UPR. To this end, we used a previously established CHO-K1 reporter cell line stably expressing XBP1s:: Turquoise and CHOP::GFP (reporting on the PERK pathway) (24). In this background, derivative *Slc33a1*-deleted single clones were generated by CRISPR-Cas9 mutagenesis and a clone, *Slc33a1*[clnA.1], was selected for further studies (Figs 2D and S2B). The loss of SLC33A1 in the IRE1/PERK dual UPR reporter cell line confirmed the significant derepression of IRE1 signalling, by both flow cytometry (Fig 2D) and increased spliced form of the endogenous XBP1 mRNA (XBP1[S]) (Fig S4A), consistent with our findings in the IRE1/ATF6α dual UPR reporter cell line. Loss of SLC33A1 also resulted in constitutive activation of PERK signalling under basal conditions (Fig 2D), a phenomenon observed in another independently derived *Slc33a1*-deleted clone from the IRE1/PERK reporter cell line (Fig S4B, *Slc33a1*Δ[clnE]). Furthermore, the loss of SLC33A1 preserved PERK signalling responsiveness to ER stress induced by tunicamycin (Fig S4C). Given that IRE1 signalling has been shown to sustain PERK expression (25), we next assessed whether the constitutive activation of IRE1 could be a contributing factor to the conspicuous up-regulation of PERK activity. Indeed, treatment with the selective IRE1 inhibitor 4µ8C restored PERK signalling near to the basal levels (Fig 2E). These results indicate that the elevated PERK activity observed is, at least in part, a downstream consequence of sustained IRE1 signalling.

## Depletion of SLC33A1 impairs ATF6α processing

The data so far point to a consistent role of SLC33A1 in UPR signalling to attenuate ATF6α and enhance IRE1 activity. However, the potential for cross-pathway negative feedback in the UPR (9, 26, 27) leaves open the question of epistasis between the *Slc33a1* mutation and the alterations in UPR signalling. To address this, we investigated the impact of SLC33A1 on ATF6α signalling in further detail. Considering the critical role of ATF6α processing by site-1 (S1P) and site-2 (S2P) Golgi proteases for the generation and trafficking of the active N-terminal domain of ATF6α (N-ATF6α) from the Golgi apparatus to the nucleus (7), we next investigated the impact of the loss of SLC33A1 on ATF6α processing by immunoblotting. To this end, we turned to a previously characterised ATF6α knock-in clone, in which the protein encoded by the endogenous *Atf6α* locus of CHO-K1 cells was tagged with 3×FLAG-mGreen Lantern (mGL) (Fig 3A, upper panel) (9). In this background, we targeted the *Slc33a1* locus and identified a mutant clone (*Slc33a1*Δ[clnA9]) that had early frameshifts in both alleles and a phenotype of basally elevated endogenous XBP1 splicing (Fig 3A, lower panel, and Fig S2B), consistent with that observed in previously characterised *Slc33a1* mutant clones (see Figs 2B and S4A).

The knock-in *Slc33a1*Δ[clnA9] cells were treated with the mild ER stress–causing agent 2-deoxy-D-glucose for various time intervals, and 3×FLAG-mGL-ATF6α was purified using GFP-Trap Agarose (ChromoTek), followed by anti-FLAG immunoblotting. Four distinct bands were observed, likely because of the presence of two methionine residues that can serve as alternative translation initiation sites in the tagged mRNA encoded by the knock-in *Atf6α* allele (Fig 3A, upper panel). This results in the production of two precursors ATF6α isoforms (ATF6[P]) with different lengths and their corresponding two processed N-terminal ATF6α fragments (ATF6[N]). The ratio of processed N-ATF6α forms (ATF6[N]) to total ATF6α was quantified to assess ATF6α processing under ER stress (Fig 3B). *Slc33a1*-deleted cells exhibited consistently reduced levels of the processed N-ATF6α form upon ER stress, compared with the parental cells. These findings align with our previous flow cytometry observations and suggest that SCL33A1 positively contributes to ATF6α processing.

**Figure 2. Depletion of SLC33A1 activates IRE1 and attenuates ATF6α signalling.**
**(A)** Left: two-dimensional contour plots of the XBP1s::mCherry and BiP::GFP signals in parental dual UPR reporter cells and a derivative *Slc33a1*-deleted clone (*Slc33a1*Δ[clnA]), under basal conditions (UT, untreated) (parental in grey; *Slc33a1*Δ[clnA] in light green) and after ER stress induction with tunicamycin (Tm, 2.5 µg/ml, 6 h) (parental in orange; *Slc33a1*Δ[clnA] in dark green). Right: quantification of the median fluorescence intensity (MFI) of XBP1s::mCherry and BiP::GFP in parental and *Slc33a1*Δ[clnA]. Bars represent the mean ± SD, and the values of the three replicates as black dots (*$P < 0.05$ and ***$P < 0.001$, two-sided unpaired Welch's *t* test). **(B)** Left: representative RT–PCR agarose gel showing XBP1 isoforms in parental and *Slc33a1*Δ[clnA] cells under basal conditions. The migration of the unspliced (XBP1[U]), spliced (XBP1[S]), and hybrid (XBP1[Hybrid], containing one strand of XBP1[U] and one strand of XBP1[S] (22)) stained DNA fragments is indicated. Right: plot of the percentage of XBP1[U] and XBP1[S] fractions of XBP1 mRNA in the two genotypes. Bars represent the mean ± SD, and the values of the three replicates as black dots (**$P < 0.01$; two-sided unpaired Welch's *t* test). **(C)** Left: two-dimensional contour plots, as in (A) of parental and *Slc33a1*Δ[clnA] dual-reporter cells untreated and after 48-h exposure to the IRE1 RNase inhibitor 4µ8C. Untreated: parental in grey; *Slc33a1*Δ[clnA] in light green; 4µ8C treated: parental in orange; *Slc33a1*Δ[clnA] in dark green. Right: quantification of the MFI of XBP1s::mCherry in parental and *Slc33a1*Δ[clnA]. Bars represent the mean ± SD, and the values of the three replicates as black dots (*$P < 0.05$; ns, not significant, two-sided unpaired Welch's *t* test). **(D)** Left: two-dimensional contour plots depicting XBP1s::mTurquoise and CHOP::GFP from parental IRE1/PERK dual UPR reporter parental cells (grey) and a derivative *Slc33a1*-deleted clone (*Slc33a1*Δ[clnA.1] in green) under basal conditions. Right: quantification of the MFI of XBP1s::mCherry and CHOP::GFP in parental and *Slc33a1*Δ[clnA.1]. Bars represent the mean ± SD, and the values of the three replicates as black dots (**$P < 0.01$ and ***$P < 0.001$, two-sided unpaired Welch's *t* test). **(E)** Left panel: two-dimensional contour plots, as in (C) of parental and *Slc33a1*Δ[clnA.1] dual-reporter cells untreated and after 48-h exposure to 4µ8C. Untreated: parental in grey; *Slc33a1*Δ[clnA.1] in dark green; 4µ8C treated: parental in orange; *Slc33a1*Δ[clnA.1] in cyan. Right: quantification of the MFI of XBP1s::mCherry and CHOP::GFP in parental and *Slc33a1*Δ[clnA.1]. Bars represent the mean ± SD, and the values of the three replicates as black dots (***$P < 0.001$; two-sided unpaired Welch's *t* test). Source data are available for this figure.

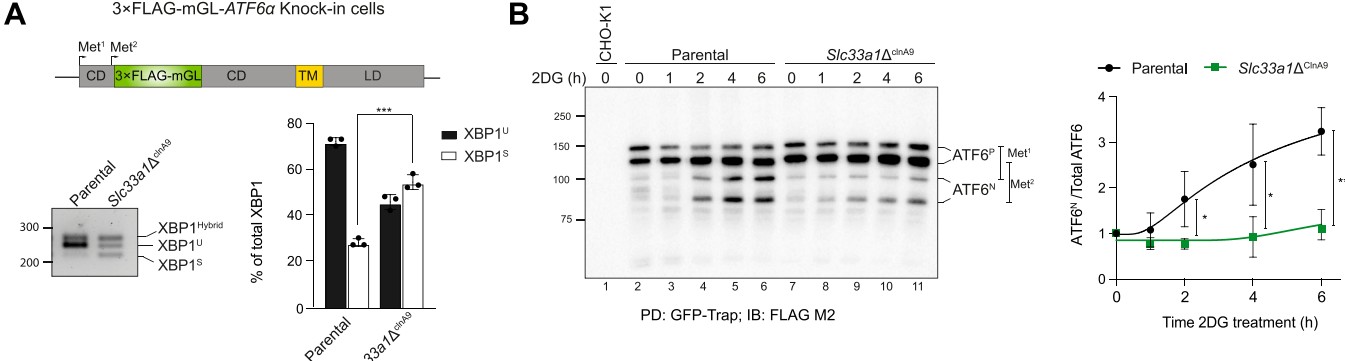

**Figure 3. SLC33A1 deletion impairs ATF6α processing.**
**(A)** Top: schema of the genetically modified endogenous CHO-K1 *Atf6α* locus, encoding ATF6α tagged with a 3×FLAG-mGreen Lantern (mGL) upstream of the second methionine (Met) start codon. The first methionine in exon 1 (Met[1]) is also marked. Colour coding: 3×FLAG-mGL (green), cytosolic domain (CD) (grey), transmembrane domain (TM) (yellow), and luminal domain (LD) (grey). Bottom: representative agarose gel showing XBP1 cDNA isoforms detected via RT–PCR in 3×FLAG-mGL-ATF6α knock-in parental cells and *Slc33a1Δ*[clnA9] cells under basal conditions. The migration of the unspliced (XBP1[U]), spliced (XBP1[S]), and hybrid (XBP1[Hybrid], containing one strand of XBP1[U] and one strand of XBP1[S] (22)) stained DNA fragments is indicated. Plot of the percentage of XBP1[U] and XBP1[S] fractions of XBP1 mRNA in the two genotypes. Bars represent the mean ± SD, and the values of the three replicates as black dots (**$P < 0.01$; two-sided unpaired Welch's *t* test). **(B)** Left: anti-FLAG M2 immunoblot of 3×FLAG-mGL-tagged ATF6α in lysates of parental and mutant *Slc33a1Δ*[clnA9] cells exposed to 2-deoxy-D-glucose (2DG, to induce ER stress) for the indicated period of time. The two forms of full-length ATF6α precursor (ATF6[P]) and two processed N-terminal ATF6α forms (ATF6[N]), arising from alternative translational initiation sites (Met[1] or Met[2]), are indicated. Parental CHO-K1 untagged cells (line 1) were used as a control for background. The immunoblot is representative of three experiments. Right: ratio of the signal from both processed ATF6[N] forms to the total ATF6α signal (ATF6[N] + ATF6[P]) for each genotype. Data are presented as the mean ± SD from three independent experiments. *$P < 0.05$ and **$P < 0.01$, according to two-way ANOVA test followed by a Bonferroni post hoc test. PD indicates pull-down, whereas IB refers to immunoblot. Source data are available for this figure.

## Constitutive Golgi localisation of ATF6α in *Slc33a1Δ* cells

Stress-dependent translocation of the membrane-bound ATF6α precursor from the ER to the Golgi is a prerequisite for its pro-teolytic activation (28). To determine whether defective ATF6α processing in cells lacking SLC33A1 correlates with defective trafficking, we used 3×FLAG-mGL-ATF6α knock-in parental cells and their *Slc33a1*-inactivated counterpart (*Slc33a1Δ*[clnA9]) for live-cell microscopy (Fig 4A). Under ER stress, parental cells exhibited a conspicuous increase in nuclear mGL-ATF6α signal compared with basal conditions that was absent in stressed *Slc33a1Δ*[clnA9] cells (Fig 4B and C). This observation aligned with the reduced pro-cessed N-ATF6α form observed in *Slc33a1Δ* cells on immunoblot (Fig 3B).

Interestingly, the defect in ATF6α processing and nuclear ac-cumulation of the processed N-terminal 3×FLAG-mGL-ATF6α in *Slc33a1Δ*[clnA9] cells was not associated with retained signal in the ER membrane, but rather with an enhanced punctate signal that co-localised (or overlapped) with a Golgi marker (Fig 4B, asterisks). This feature suggests that SLC33A1 acts at a downstream step after ATF6α trafficking to the Golgi.

Resistance to proteolytic cleavage by S1P and S2P uncouples ER-to-Golgi trafficking from the downstream processing of ATF6α, facilitating interrogation of the two events separately. Therefore, we used a previously described CHO-K1 cell stably expressing a GFP-tagged version of cgATF6α_LD (luminal domain), with muta-tions in the S1P and S2P cleavage sites (GFP-ATF6α_LD[S1P,S2Pmut]) (9) to follow up on the suggestion of a perturbation to ATF6α's itin-erary in cells lacking SLC33A1 (Fig 4D). A derivative *Slc33a1*-deleted clonal cell (called *Slc33a1Δ*[clnH]) was generated for analysis (Fig S2B). Under both basal and stressed conditions, the GFP-

ATF6α_LD[S1P,S2Pmut] signal in the *Slc33a1* mutant cells was distinctly punctate with significant co-localisation to a Golgi marker (Fig 4E and F). In contrast, parental cells exhibited a diffuse GFP-ATF6α signal under basal conditions, with noticeable co-localisation of GFP-ATF6α with the Golgi marker only upon exposure to ER stress (Fig 4E and F). This observation was reproduced in a second *Slc33a1*-deficient GFP-ATF6α_LD[S1P,S2Pmut] clone (Fig S5A and B, *Slc33a1Δ*[clnK]).

The punctate Golgi-associated localisation of the fluorescent ATF6α fusion proteins under basal conditions is consistent with disruption of an ATF6α-driven feedback loop in the *Slc33a1Δ*[clnA9] and *Slc33a1Δ*[clnH] cells that functions homeostatically, even in the absence of pharmacologically imposed stress. These findings suggest that in the absence of SLC33A1, ATF6α constitutively translocates to the Golgi and once there encounters a block in its downstream processing.

## Impaired Golgi-type sugar modification of ATF6α in *Slc33a1Δ* cells

Upon arrival, the glycans on the luminal domain of the ATF6α precursor become substrates for Golgi-localised enzymes. Given the evidence above for perturbation of a Golgi-localised node in ATF6α's itinerary and the implication of SLC33A1 in providing the acetyl-CoA building blocks for post-ER glycan modification, we wished to compare the maturation of ATF6α glycans in parental and *Slc33a1*-mutant cells.

ATF6 contains three conserved N-linked glycosylation sites in its ER luminal domain, and its N-glycosylation state has been sug-gested to influence its trafficking (8). O-linked glycosylation of ATF6 has also been reported (6), although the functional impli-cations of these Golgi-modified sugars remain unclear. To reveal

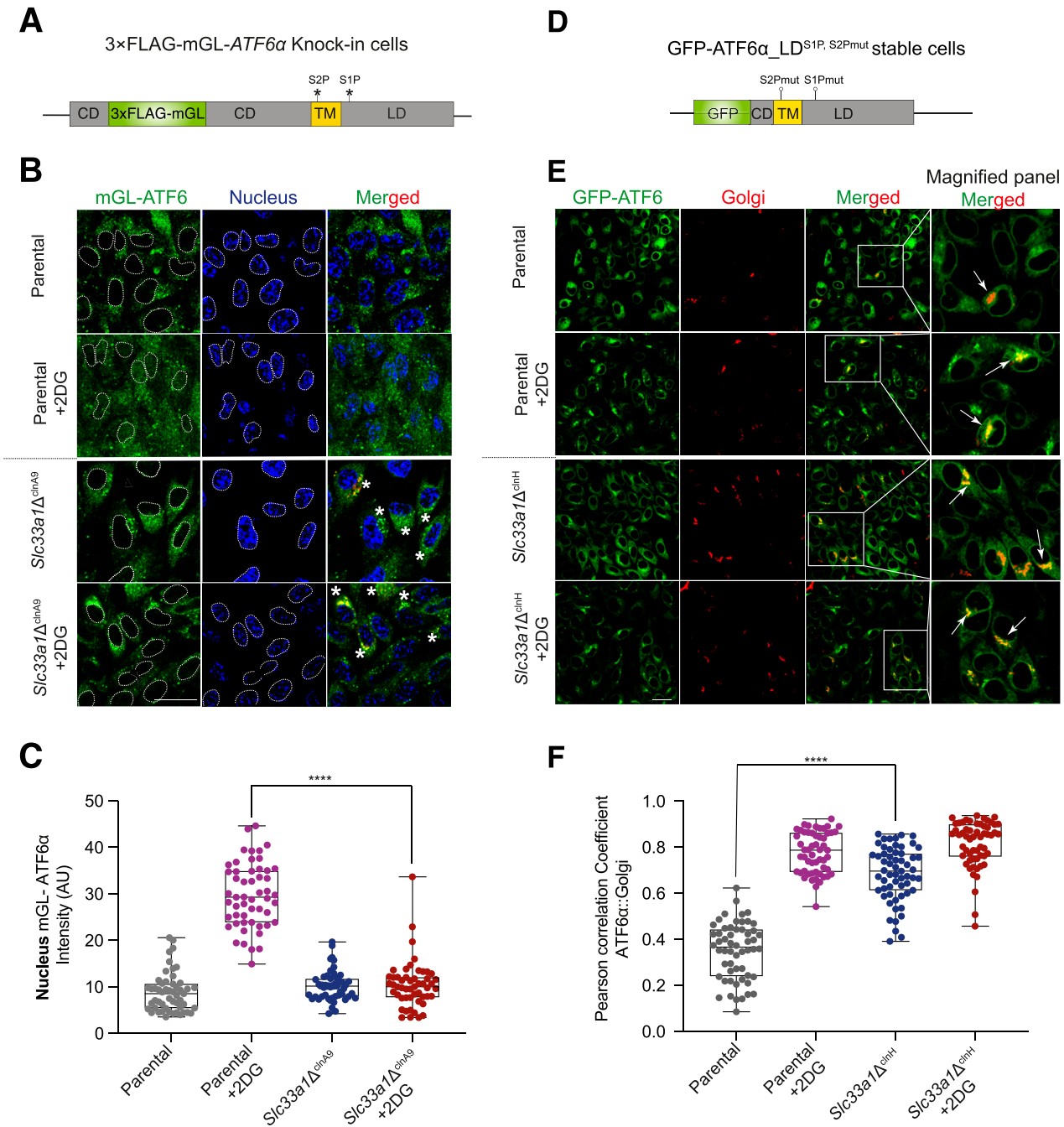

**Figure 4. Constitutive Golgi localisation of ATF6α in *Slc33a1*Δ CHO cells.**
**(A)** Schema of the genetically modified endogenous *Atf6α* locus (as in Fig 3A). The S1P and S2P cleavage sites are indicated by stars. **(B)** Representative live-cell confocal microscopy images of 3×FLAG-mGL-ATF6α knock-in parental cells and *Slc33a1*Δ<sup>clnA9</sup> derivative clone under basal and ER stress conditions (+2DG). mGL-ATF6α is shown in green, and the nucleus is in blue. Cells were transiently transfected with the pmScarlet_Giantin-C1 plasmid to label the Golgi apparatus (see the merged panel). Co-localisation of mGL-ATF6α to a punctate pattern (Golgi-like, yellow) is indicated by asterisks in the merged panel. Scale bar: 20 μm. **(C)** Quantification of mGL-ATF6α signal intensity in the nucleus (outlined by dashed lines), measured using ImageJ software from images in panel (B). Data from parental cells (n > 50) and *Slc33a1*Δ<sup>clnA9</sup> cells (n > 50) are shown as a box-and-whisker plot, displaying all individual values, including minimum and maximum intensities. Statistical significance was determined using two-sided unpaired Welch's *t* test (****$P < 0.0001$). **(D)** Schema of the stable transgene encoding an N-terminally tagged ATF6α derivative (as in (A) above) in which GFP replaces most of the cytosolic domain, and mutations in the S1P and S2P cleavage sites preclude activating proteolysis and favour Golgi retention of the trafficked protein. Colour coding: GFP tag (green), cytosolic domain (CD, grey), transmembrane domain (TM, yellow), and luminal domain (LD, grey). **(E)** Representative live-cell confocal microscopy images as in (B) above. Cells were transiently transfected with pmScarlet_Giantin-C1 to visualise the Golgi (in red). Insets show magnified regions, with arrows highlighting GFP-ATF6α co-localisation with the Giantin Golgi marker. Scale bar: 20 μm. **(F)** Pearson's correlation coefficients were used to quantify co-localisation of GFP-ATF6α_LD<sup>S1P,S2Pmut</sup> with Giantin in parental cells (n > 50) and *Slc33a1*Δ<sup>clnH</sup> cells (n > 50). Data are presented as a box-and-whisker plot, including all individual values along with minimum and maximum intensities. Co-localisation analysis was performed using Volocity software, and statistical significance was determined using two-sided unpaired Welch's *t* test (****$P < 0.0001$).

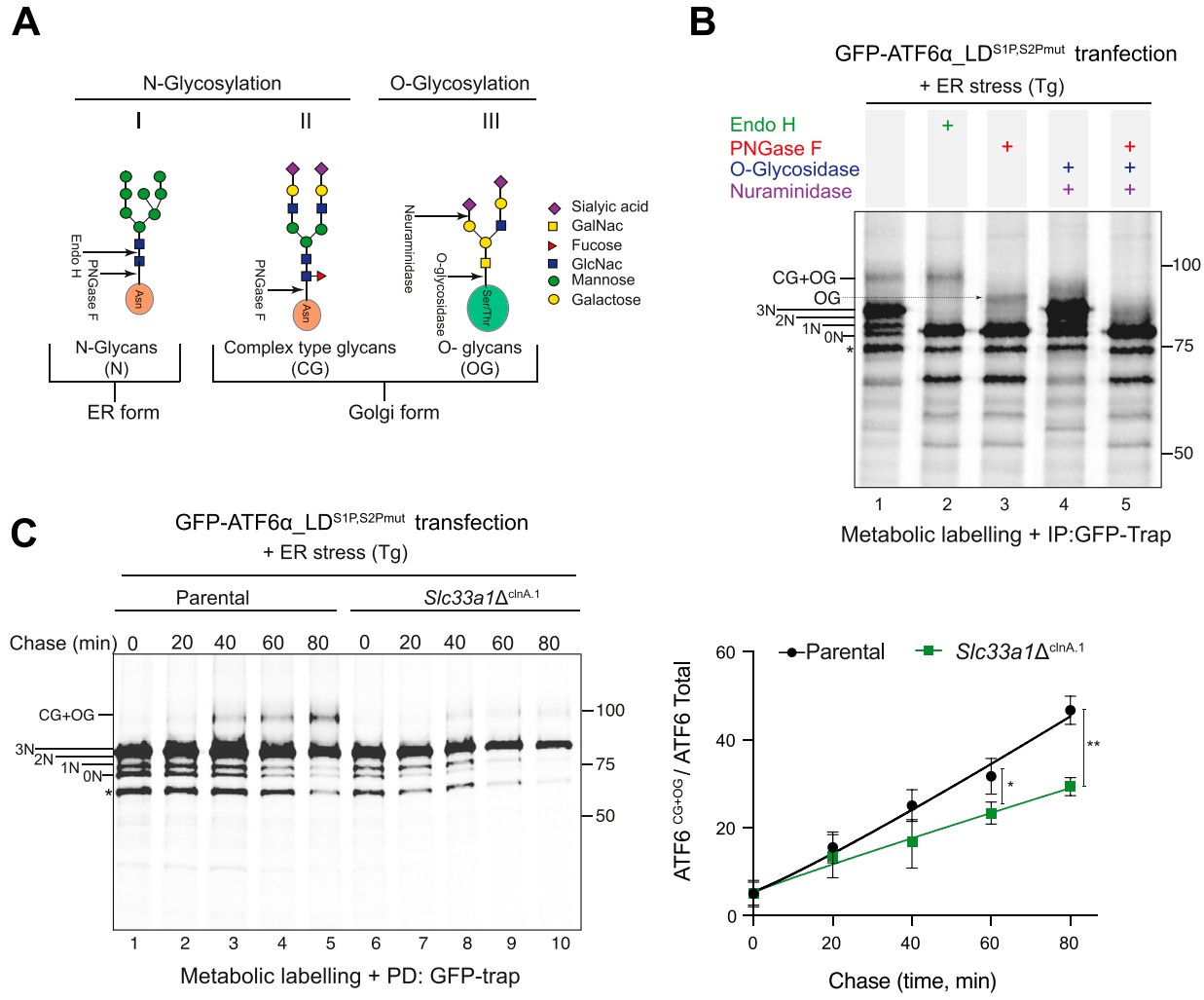

**Figure 5. Impaired Golgi-type sugar modification of ATF6α in *Slc33a1*Δ cells.**
**(A)** Schema of glycan modifications in the ER and Golgi and their sensitivity to glycosidases. **(B)** Autoradiograph of GFP-ATF6α_LD[S1P,S2Pmut] immunoprecipitated from cell lysates of [35]S-Met/Cys-metabolically labelled transfected CHO-K1 cells. To induce ATF6α trafficking from ER to Golgi, cells were treated with thapsigargin (Tg, 300 nM). Samples were exposed to the indicated glycosidases. The bands corresponding to ATF6α-LD (luminal domain) with one (1N), two (2N), or three (3N) N-glycan modifications are indicated. O-glycan and complex-type glycans are annotated as OG and CG, respectively. (*) denotes a non-specific band. **(C)** Left: autoradiograph as in (B) of samples from a pulse-chase experiment in cells of the indicated genotype. Cells were treated with Tg for 30 min before the pulse, and Tg treatment was maintained during the [35]S Met/Cys pulse labelling (15 min) and the chase times (20–80 min). Shown is a representative of three experiments. Right: ratio of the signal from complex-type (CG) and O-glycan (OG)–modified GFP-ATF6α-LD[S1P,S2Pmut] to the total ATF6 signal for each genotype over the chase period. Data are presented as the mean ± SD, n = 3, *P < 0.05 and **P < 0.01, according to two-way ANOVA test followed by a Bonferroni post hoc test.
Source data are available for this figure.

the palette of ATF6 glycosylation states, we used the GFP-tagged version of cgATF6α_LD (lacking the S1P and S2P cleavage sites; GFP-ATF6α_LD[S1P,S2Pmut]) in CHO-K1 to analyse the mobility of GFP-tagged proteins by autoradiography on SDS–PAGE of [35]S-metabolically labelled material immunopurified from stressed cells. Cell lysates were subjected to enzymatic digestion with different glycosidases to characterise ER and Golgi glycosylation modifications (Fig 5A and B). In the absence of enzymatic digestion, four distinct ATF6α-specific bands appeared (Fig 5B, Lane 1). The four lower molecular weight bands were labelled as 0N–3N, denoting the number of N-glycans attached to ATF6α. A

fifth, higher molecular weight band was interpreted as a mixture of complex-type glycan (CG) and O-glycan (OG) forms of ATF6α (labelled CG+OG), typically generated in the Golgi. As expected, upon treatment with Endo H (Fig 5B, Lane 2), the N3, N2, and N1 bands collapsed into a single, faster migrating band, indicating that these represent simple N-glycans characteristic of the ER-localised form of ATF6α. The upper CG+OG band remained Endo H–resistant, consistent with the presence of Golgi-modified glycans. When treated with PNGase F alone, which cleaves both N-glycan and CG forms (Fig 5B, Lane 3), N3, N2, and N1 collapsed to N0 (as expected). The upper CG+OG band shifted to a slightly

slower migrating species (labelled as OG), suggesting a mixed composition of N- and O-glycans. This was confirmed by its disappearance from samples co-treated with O-glycosidases (Fig 5B, lanes 4 and 5).

With Fig 5B as a reference, we compared the time-dependent changes in glycosylation of the GFP-ATF6$\alpha$_LD$^{S1P,S2Pmut}$ sentinel in parental cells and the Slc33a1$\Delta^{clnA.1}$ derivative clone by pulse-chase labelling under ER stress conditions. In parental cells, there was a progressive increase in the CG+OG Golgi-modified ATF6$\alpha$ population over the chase period (Fig 5C, lanes 3–5). Interestingly, in Slc33a1$\Delta$ cells, the formation of these Golgi-modified populations was significantly slower (Fig 5C, lanes 8–10).

Taken together, our findings point to an uncoupling between trafficking to the Golgi, which is enhanced in Slc33a1 mutant cells, and Golgi modification of glycans and proteolytic processing of ATF6$\alpha$ by S1P and S2P Golgi proteases, which is diminished by the depletion of SLC33A1. These features are associated with attenuated subsequent translocation of the cleaved N terminus to the nucleus and ultimately ATF6$\alpha$ signalling.

O-acetylation of sialic acids is a post-translational modification that modulates the stability, trafficking, and function of secreted glycoproteins (29). In the Golgi, acetyl groups from acetyl-CoA are transferred to sialic acid residues at positions C4, C7, C8, or C9 on nascent glycoproteins (30). This modification protects sialic acids from sialidase-mediated cleavage, preserves glycan integrity during Golgi processing, and can influence glycoprotein sorting by altering recognition by lectins and cargo receptors. Because SLC33A1 has been reported to play a role in ganglioside formation by supplying acetyl-CoA to the Golgi for O-acetylation, we sought to compare O-acetyl modification of ATF6$\alpha$ in WT and Slc33a1-deleted cells by mass spectrometry. Although we successfully identified terminal sialic acid modifications on the N-glycans of ATF6$\alpha$, we were unable to detect their O-acetylated forms, likely because of the inherent technical limitations of mass spectrometry in capturing these labile modifications, which are prone to loss during ionisation and fragmentation. However, we observed a marked increase in the unmodified sialylated N-glycans, the precursors to O-acetylated forms, in the Slc33a1-deleted cells compared with the parental cell line. This finding is consistent with accumulation of sialylated glycans that remain unacetylated because of defective glycan processing in the absence of SLC33A1 (Fig S6A–C).

Based on these previous results, we were prompted to investigate the role of CASD1 (CAS1 Domain Sialic Acid O-Acetyltransferase), the main Golgi-resident sialic acid O-acetyltransferase that uses acetyl-CoA to generate 9-O-acetylated sialic acids on glycans and gangliosides (31), as a potential downstream effector of SLC33A1. Given that CASD1 catalyses the transfer of acetyl groups from acetyl-CoA to the hydroxyl group at the C-7 or C-9 position of CMP-activated sialic acid (CMP-Neu5Ac), we sought to evaluate whether the loss of CASD1 phenocopies aspects of Slc33a1 deletion. To test this, the Casd1 gene was disrupted by CRISPR-Cas9 gene editing in IRE1/ATF6$\alpha$ dual UPR reporter cells, generating derivative Casd1-deleted clones (Fig S7A). Although loss of CASD1 produced a more modest phenotype than the Slc33a1-deficient state, it followed the overall trend. Specifically, Casd1 loss led to a modest yet significant basal derepression of the IRE1 pathway, similar to that observed in Slc33a1-depleted cells, along with a reduced induction of the ATF6$\alpha$ reporter (BiP::GFP) under ER stress conditions induced by 2DG (Fig S8A and B). Together, these findings suggest that CASD1 contributes to optimal ATF6$\alpha$ signalling capacity at the Golgi and may act downstream of SLC33A1, rather than directly regulating basal ER stress pathways.

## Depletion of NAT8 acetyl-CoA acetyltransferases does not phenocopy SLC33A1 loss

Within the ER lumen, acetyl-CoA has been proposed to serve as a substrate for the ER membrane–bound lysine acetyltransferases of the NAT8 family of enzymes (32) (Fig 6A). These acetyltransferases transfer the acetyl group from acetyl-CoA to N$^\varepsilon$-lysine of various protein substrates, a process that is thought to affect protein secretion, molecular stabilisation, conformational protein assembly, and overall cellular homeostasis (33, 34, 35). To determine whether defective protein N$^\varepsilon$-lysine acetylation contributes to the phenotype observed in Slc33a1-deleted cells, we measured the consequences of Nat8 inactivation on ATF6$\alpha$ signalling.

CHO-K1 cells express two isoforms of the Nat8 gene: Nat8 and Nat8b, redundancy among which could explain failure to detect them in genome-wide screens. Both isoforms were targeted independently by CRISPR-Cas9 gene editing in IRE1/ATF6$\alpha$ dual UPR reporter cell lines to generate derivative Nat8-deleted clones (Fig S7B and C). The consequences of Nat8 depletion were assessed by measuring reporter activity via flow cytometry. Neither the targeting Nat8 isoforms individually nor dual inactivation of both isoforms replicated the features of Slc33a1-deleted cells. Instead, inactivation of either Nat8 or Nat8b or both resulted in elevated reporter activity of both XBP1::mCherry and BiP::GFP (Figs 6B and S9). Together, these findings support a role of N$^\varepsilon$-lysine acetylation in maintaining ER protein homeostasis (as suggested previously, references 34, 35) but indicate that Nat8 and Slc33a1 influence ER homeostasis and ATF6$\alpha$ signalling through distinct mechanisms.

## Discussion

This unbiased genome-wide CRISPR-Cas9 screen establishes a link between the ER-localised transmembrane solute transporter SLC33A1 and ATF6$\alpha$ activation. Initially established by tracking the activity of a UPR pathway reporter, the role of Slc33a1 as a positive and selective regulator of ATF6$\alpha$ activity was confirmed by gene deletion and examining endogenous UPR markers. The defect caused by the deletion of Slc33a1 mapped to a post-ER step in the itinerary of ATF6$\alpha$ processing. It correlated with a defect in activating proteolysis of the ATF6$\alpha$ precursor and in post-ER processing of its glycans, despite its constitutive trafficking from the

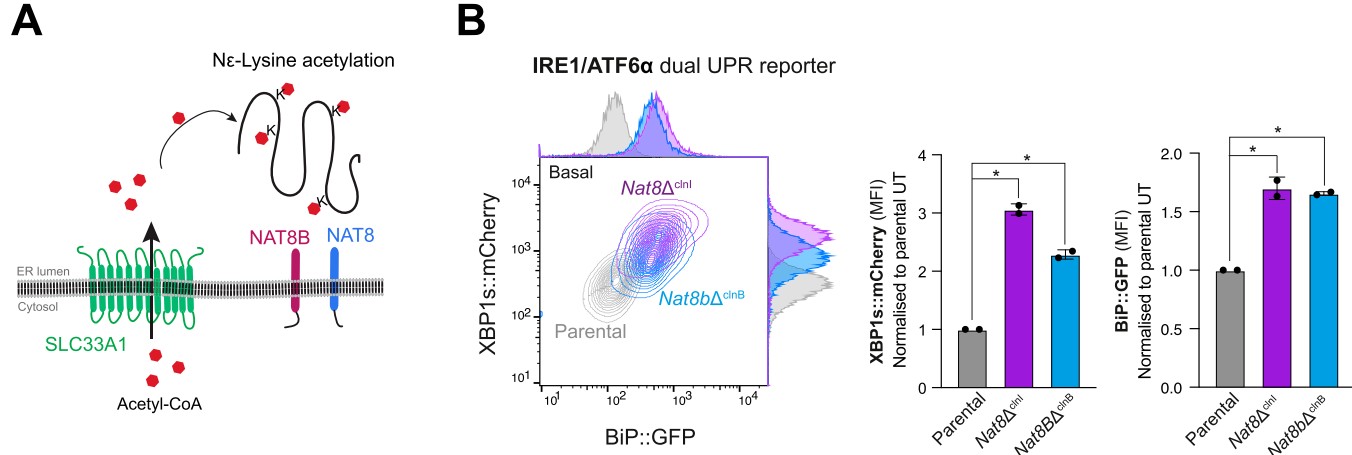

**Figure 6.   Depletion of ER-localised acetyltransferases does not phenocopy SLC33A1 loss.**
**(A)** Schematic representation of N$^\varepsilon$-lysine acetylation in the ER. Acetyl-CoA (red hexagons), transported into the ER lumen of the secretory pathway by SLC33A1, serves as an acetyl donor for NAT8 and NAT8B, two luminal acetyltransferases that modify lysine residues on protein substrates, resulting in N$^\varepsilon$-lysine acetylation. **(B)** Left panel: two-dimensional contour plots depicting XBP1s::mCherry and BiP::GFP signals in parental IRE1/ATF6$\alpha$ dual UPR reporter cells (grey) and derivative *Nat8*-deleted clones (*Nat8Δ*$^{clnl}$: purple; *Nat8bΔ*$^{clnB}$: blue). Right panel: quantification of the median fluorescence intensity (MFI) of XBP1s::mCherry and BiP::GFP from two independent experiments is shown in bar diagrams representing the mean ± SD. Statistical analysis was performed using two-sided unpaired Welch's *t* test (**P* < 0.05).

ER to the Golgi. These findings point to an unanticipated role of ER metabolite(s) in promoting ATF6$\alpha$ signalling.

*Slc33a1*'s deletion has discordant effects on UPR pathways: promoting IRE1 activity while inhibiting ATF6$\alpha$ activity. This pattern could reflect pleiotropic effects of SLC33A1 on distinct branches of the UPR (and more broadly on cellular adaptations to stress (36)), but it is insufficient to discriminate primary from secondary effects. However, considering the known cross-talk between UPR branches, the impaired processing of Golgi-localised ATF6$\alpha$ offers a parsimonious explanation for the observations reported here: ATF6$\alpha$'s dominant role in regulating chaperone gene expression in the vertebrate UPR (37, 38) specifies disruption of ER proteostasis, derepressing both IRE1 signalling and the upstream-most steps in ATF6$\alpha$ activation in *Slc33a1* mutant cells. This model is consistent with the enhanced splicing of XBP1 and the constitutive trafficking of the ATF6$\alpha$ precursor to the Golgi observed in the *Slc33a1* mutant cells here and with the recent observation that a chemical inhibitor of SLC33A1 selectively activates the IRE1 pathway in HEK293 cells (39 *Preprint*). The smooth transit of ATF6$\alpha$ from the ER to the Golgi is consistent with findings that point to normal ER-to-Golgi trafficking of a model secretory protein in *Slc33a1* mutant mouse cells (15). However, the block in ATF6$\alpha$ processing once arrived in the Golgi represents an unanticipated finding.

The genetic analysis described here outlines a plausible picture of the epistasis between *Slc33a1*, *Atf6α*, and *Ern1* (encoding IRE1), but does not yet uncover the underlying molecular mechanism. Although phenotypic data from humans (40) and mice (15, 41) with *Slc33a1* mutations are consistent with secretory pathway dysfunction, they do not currently clarify how SLC33A1 affects ATF6$\alpha$ processing and the field is therefore wide open for speculation. It is important to acknowledge that although the evidence supporting acetyl-CoA transport by SLC33A1 is strong, there is nothing to argue against the

possibility that other acetyl-CoA–related ligands containing a 3'-phosphorylated ADP moiety, such as ATP, dATP, and ADP, which have recently been shown to be transported by SLC33A1 (11), might also contribute to defective ATF6$\alpha$ activation observed in *Slc33a1* mutant cells. This aligns with ongoing questions about how metabolites like ATP, known to play a key role in protein folding in the secretory pathway (42, 43), are imported into the ER.

Nonetheless, the prism of defective acetyl-CoA transport into the ER offers valuable insight. The divergent phenotypes of *Nat8* and *Slc33a1* inactivation argue against a central role of N$\varepsilon$-lysine acetylation in the observed ATF6$\alpha$ activation defect, though this must be interpreted with caution because of potential differences in mutation strength and pathway-specific pleiotropy.

O-acetylation of terminal sialic acid residues on N-glycans, which relies on a supply of acetyl-CoA as a donor, is believed to be a trans-Golgi modification (44), whereas activating processing of the ATF6$\alpha$ precursor is believed to occur in the cis-Golgi (45). Nonetheless, it remains possible that a so far unrecognised acetyl-CoA–dependent modification promotes ATF6$\alpha$ processing. Alternatively, given the evidence for a broad defect in secretion in *Slc33a1*-compromised cells, it is equally plausible that a deficiency in acetyl-CoA import (or some other metabolite) results in a widespread impairment of Golgi function, which, among other consequences, disrupts ATF6$\alpha$ activation (Fig 7). Although our data indicate that SLC33A1-dependent metabolite import is required for proper ATF6$\alpha$ maturation, the substrate(s) responsible remain(s) uncertain. Our evidence that SLC33A1 transports acetyl-CoA is indirect, and alternative metabolites, such as oxidised glutathione, have been recently proposed and could also contribute to the observed ATF6$\alpha$ processing defects (46 *Preprint*).

Whether defective ATF6$\alpha$ activation contributes to the clinical manifestations of *Slc33a1* mutations remains to be determined.

<center>Wild-type conditions *Slc33a1*-deleted conditions</center>

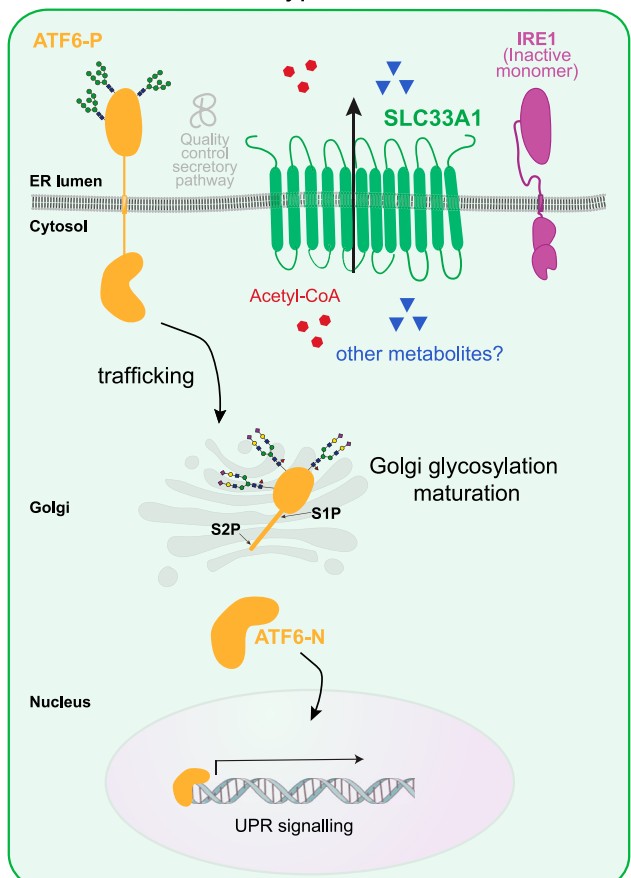
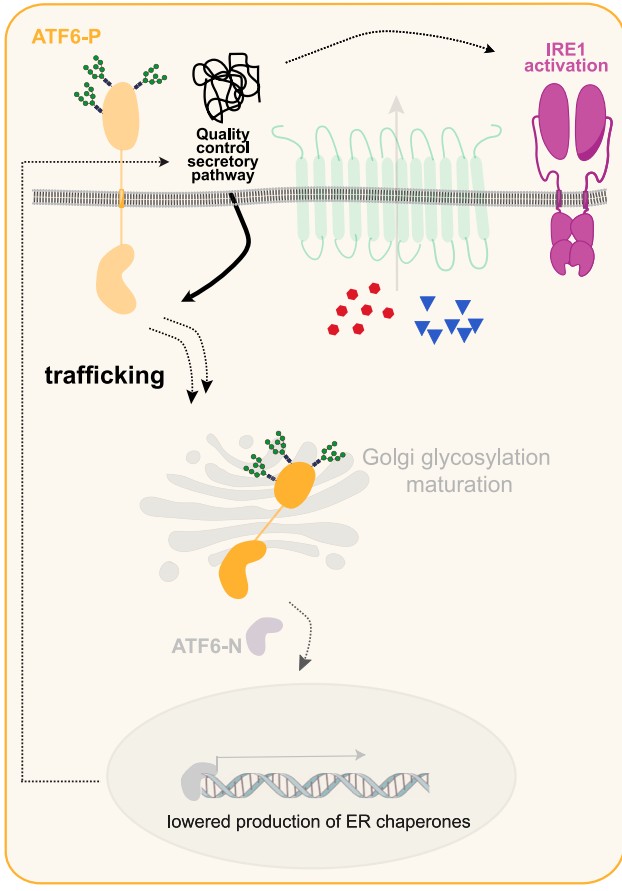

**Figure 7.  Proposed model.**
SLC33A1, a multipass transmembrane protein located at the ER, transports acetyl-CoA from the cytosol into the ER, to serve as a substrate for acetylation reactions in the secretory pathway. Changes in metabolite abundance caused by loss of SLC33A1 (right panel) impair the processing of the transmembrane ATF6 precursor (ATF6-P) in the Golgi and the liberation of its active fragment, ATF6-N, by the site-specific proteases S1P and S2P, a defect that correlates with altered glycosylation of ATF6-P. Lower ATF6$\alpha$ signalling in the nucleus reduces production of ER chaperones and subsequently impairs protein folding homeostasis in the ER. The resulting enhanced burden of unfolded proteins explains both the heightened IRE1 activity and the constitutive trafficking of ATF6 to the Golgi observed in cells lacking *Slc33a1*. SLC33A1 activity may impinge on other aspects of the cellular function. Although SLC33A1 may also affect other cellular processes or transport additional metabolites, the chain of events uncovered here attributes a selective role to metabolites transported into the secretory pathway in ATF6 activation.

However, given emerging links between intermediary metabolism, ageing, and proteostasis in the secretory pathway, this work underscores SLC33A1 as a key genetic entry point for dissecting how metabolic state regulates the unfolded protein response and serves as a stimulus for further research.

# Materials and Methods

Detailed methods are provided in the supplemental materials of this article, including Table S1.

### Mammalian cell culture

Adherent Chinese hamster ovary (CHO-K1) cells (ATCC CCL-61) were cultured in Ham's F12 nutrient mixture (Sigma-Aldrich), supplemented with 10% (vol/vol) serum (FetalClone-2, HyClone), 2 mM L-glutamine (Sigma-Aldrich), and 1% penicillin/streptomycin (Sigma-Aldrich). The cells were incubated at 37°C in a humidified environment containing 5% $CO_2$. As specified, cells were exposed to tunicamycin (Tm, 2.5 $\mu$g/ml), 2-deoxy-D-glucose (2DG, 4 mM), 4$\mu$8C (4 $\mu$M), dithiothreitol (DTT, 2 mM), or thapsigargin (Tg, 300 nM) for varying times depending on the specific experiments as indicated. Untreated cells were treated with DMSO solvent vehicle control. Transfections in CHO-K1 cells were performed using Lipofectamine LTX (Thermo Fisher Scientific) at 1:3 DNA ($\mu$g)-to-LTX ($\mu$l) ratio.

### Flow cytometry analysis

To assess UPR reporter activity, cells were seeded in six-well plates and grown to ~80–90% confluency before being treated with the specified compounds for the indicated period. For flow cytometry analysis, cells were washed twice with PBS, then collected in PBS

containing 4 mM EDTA, and analysed using a BD LSRFortessa flow cytometer (BD Biosciences), with 20,000 cells processed per sample. For FACS sorting, cells were harvested in PBS containing 4 mM EDTA and 0.5% BSA and sorted using a Beckman Coulter MoFlo cell sorter. Sorted cells were either collected in fresh culture medium containing 20% (vol/vol) serum (FetalClone-2; HyClone), as bulk populations, or individually deposited into 96-well plates for subsequent expansion. Live cells were gated based on FSC-A/SSC-A profiles, whereas singlets were gated using FSC-W/SSC-A. Reporter fluorescence was detected as follows: BiP::GFP using a 488-nm excitation laser and a 530/30-nm emission filter; XBP1s::mCherry using a 561-nm excitation laser and a 610/20-nm emission filter; and mTurquoise and BFP using a 405-nm excitation laser and a 450/50-nm emission filter. FlowJo v10 (BD Biosciences) was used for data analysis, and median fluorescence intensity was further analysed and plotted using GraphPad Prism 10.

### RNA extraction and cDNA synthesis

Total RNA was extracted using TRIzol reagent (Invitrogen), with cells incubated in the reagent for 10 min before transferring to clean tubes. To each sample, 200 $\mu$l of chloroform was added, followed by vigorous vortexing for 1 min and centrifugation at 13,500$g$ for 15 min at 4°C. The upper aqueous phase was carefully removed and mixed with an equal volume of 70% ethanol. RNA was then purified using PureLink RNA Mini Kit (Invitrogen) according to the manufacturer's protocol and eluted for downstream use. RNA concentrations were measured using a NanoDrop spectrophotometer, and the integrity of 2 $\mu$g RNA from each sample was checked on a 1.2% agarose gel.

For reverse transcription–PCR (RT–PCR), 2 $\mu$g of total RNA was denatured at 70°C for 10 min in a reverse transcription buffer (Cat #EP0441; Thermo Fisher Scientific) containing 0.5 mM dNTPs and 0.05 mM Oligo(dT)18 primers. The reaction was then supplemented with 0.5 $\mu$l of RevertAid Reverse Transcriptase (Thermo Fisher Scientific) and 100 mM DTT and incubated at 42°C for 90 min. The synthesised cDNA was diluted 1:4 for further analysis.

### PCR-based analysis of XBP1 mRNA splicing

Spliced (XBP1s) and unspliced (XBP1u) variants of XBP1 mRNA were amplified from synthesised cDNA using PCR with primers designed to flank the IRE1-mediated splicing site (hamXBP1.19S: GGCCTTGTAATTGAGAACCAGGAG and mXBP1.14AS: GAATGCC-CAAAAGGATATCAGACTC). The amplification was carried out using Q5 High-Fidelity 2X Master Mix (New England Biolabs), following the protocol described by Tung et al [9]. The PCR products corresponding to XBP1u (255 bp) and XBP1s (229 bp) were separated on a 3% agarose gel and visualised using a SYBR Safe DNA stain (Invitrogen). In addition, a hybrid amplicon of ~280 bp was detected. The proportion of XBP1s was quantified by measuring the intensity of gel bands using Fiji software (version 1.53c), with the hybrid band assumed to be equally composed of XBP1u and XBP1s.

### qRT-PCR

Quantitative real-time PCR (qRT-PCR) was performed using SYBR Green Master Mix (Applied Biosystems) on Bio-Rad CFX384 Touch Real-Time PCR Detection System. Each reaction was carried out in a final volume of 10 $\mu$l containing diluted cDNA template and gene-specific primers (see Table S1). The thermal cycling conditions were as follows: initial denaturation at 95°C for 10 min, followed by 40 cycles of denaturation at 95°C for 15 s and annealing/extension at 60°C for 1 min. A melt curve analysis was performed at the end of each run to confirm primer specificity. Gene expression levels were normalised to the signal of *Rpl27* and calculated using the ΔΔCt method. All samples were run in technical triplicates and experiments were performed with at least three independent biological replicates.

### Generation of knockout cells using CRISPR-Cas9

Single-guide RNAs (sgRNAs) targeting exon regions of *Slc33a1*, *Nat8*, and *Nat8b* in *Cricetulus griseus* were designed using CCTop-CRISPR/Cas9 target online predictor (https://cctop.cos.uni-heidelberg.de). These sgRNAs were subsequently cloned into one of the following CRISPR-Cas9 plasmid vectors: pSpCas9(BB)-2A-mCherry_V2 (UK1610) or pSpCas9(BB)-2A-mTurquoise (UK2915) as described by reference [47]. Plasmid constructs were validated by DNA Sanger sequencing to ensure correct insertion. For cell editing, previously described IRE1/ATF6$\alpha$ dual UPR reporter cells ([9]), IRE1/PERK dual UPR reporter cells (reference [24]), 3×FLAG-mGL-ATF6$\alpha$ knock-in cells ([9]), and GFP-ATF6$\alpha$_LD$^{S1P,S2Pmut}$ stable cells ([9]) were transfected with 1 $\mu$g of the sgRNA/Cas9 plasmid using Lipofectamine LTX (Thermo Fisher Scientific) in a six-well plate. After 48 h, cells with high mTurquoise or mCherry (depending on the reporter cells used to generate the knockout) fluorescence were sorted into 96-well plates at one cell per well using a MoFlo cell sorter (Beckman Coulter). Successful knockout clones were verified via Sanger sequencing (Fig S2B).

These studies were carried out in genetically malleable CHO-K1 cells. Although we acknowledge that validation in additional cell types would strengthen the study, when attempting to generate clonal *Slc33a1* knockouts in aneuploid human cell lines such as HeLa cells, we found technical problems to generate single clones. Nevertheless, UPR regulation is largely conserved supporting the broader relevance of our findings.

### IncuCyte-based cell proliferation assay

Approximately 30,000 cells of *Slc33a1* knockout lines and their corresponding parental control cells were seeded in 24-well plates. Cell proliferation was assessed by measuring cell confluency over the indicated time course using the IncuCyte S3 live-cell imaging system (Essen BioScience). Phase-contrast bright-field images were acquired using a 20× objective at 2-h intervals with an exposure time of 200 ms. Cell confluency was quantified using IncuCyte software (version 2021A).

## Live-cell confocal microscopy

Cells stably expressing GFP-ATF6α_LD[S1P,S2Pmut] or cells having endogenous *cgAtf6α* with an N-terminal 3×FLAG-mGL were cultured on 35-mm glass-bottom imaging dishes (MatTek) and pmScarlet_Giantin-C1 plasmid to label the Golgi apparatus in red. After 24 h of incubation post-transfection, cells were treated with either 4 mM 2-deoxy-D-glucose (2DG) for 3 h or 2 mM dithiothreitol (DTT) for 1 h, depending on the experimental condition.

Live imaging was performed using a Zeiss LSM 780 inverted confocal laser scanning microscope equipped with a temperature- and $CO_2$-controlled chamber (37°C, 5% $CO_2$). A ×64 oil immersion objective was used, with the pinhole set to 1 Airy unit. Fluorescent signals were captured using appropriate laser lines: 488 nm for GFP or mGL, 405 nm for DAPI, and 594 nm for mCherry. Co-localisation between different fluorescent markers within single cells was analysed using Volocity software (version 6.3; PerkinElmer) by calculating the Pearson correlation coefficient.

## Pulse-chase radiolabelling and immunoprecipitation using GFP-Trap Agarose to detect Golgi-modified glycans

Parental CHO-K1 cells and their *Slc33a1*-deleted derivatives were seeded in 12-well plates. After 18-h post-seeding, cells were transfected with the GFP-ATF6α_LD[S1P,S2Pmut] construct (UK3116). Twenty-four hours post-transfection, cells were cultured in methionine/cysteine-deficient DMEM (Cat# 21013024; Gibco) for 30 min to deplete endogenous amino acids. To initiate ATF6α trafficking from the ER to the Golgi, cells were treated with 300 nM thapsigargin (Tg) for 30 min, with Tg treatment maintained throughout the subsequent radiolabelling step. Cells were then pulse-labelled for 10 min using 5.5 $\mu$Ci/well of [35S] methionine/cysteine (Expre35S Protein Labelling Mix), followed by either immediate harvesting or a chase period of up to 80 min. Post-pulse, media were collected, and cells were washed twice with ice-cold PBS containing 0.1 mg/ml cycloheximide to halt translation. Cells were then pelleted by centrifugation at 845*g* for 5 min at 4°C and lysed using Nonidet lysis buffer (150 mM NaCl, 50 mM Tris–HCl, pH 7.5, 1% NP-40, 0.05 mM TCEP) supplemented with protease inhibitors (0.1 mg/ml cycloheximide, 1 mM PMSF, 4 $\mu$g/ml aprotinin, and 2 $\mu$g/ml pepstatin A). After lysis, post-nuclear supernatants were obtained by centrifuging the lysates at 20,000*g* for 15 min at 4°C, and the soluble portion was collected. Radiolabelled proteins were subjected to digestion with Endo H, PNGase F, O-glycosidase, and neuraminidase individually or in combination. Equal volumes of the cleared digested lysates were incubated with 15–20 $\mu$l GFP-Trap Agarose (ChromoTek) pre-equilibrated in lysis buffer. The mixture was incubated for 2 h at 4°C with rotation. The beads were then recovered by centrifugation (845*g*, 3 min) and washed four times with washing buffer (50 mM Tris–HCl, 1 mM EDTA, 0.1% Triton X-100). Proteins were eluted in 35 $\mu$l of 4×SDS-DTT sample for 10 min at 55°C, resolved on 12.5% SDS–PAGE gels, and visualised by autoradiography using a Typhoon biomolecular imager (GE Healthcare). Signal quantification was carried out using Fiji (ImageJ).

## Mass spectrometric analysis of ATF6α_LD[S1P,S2Pmut] protein glycosylation

Parental and *Slc33a1*-deleted cell lines stably expressing GFP-ATF6α_LDS[1P,S2Pmut] were cultured on four 140-mm plates. After 18 h post-seeding, cells were treated with 4 mM DTT for 1 h to induce ATF6 trafficking to the Golgi. After treatment, cells were washed with ice-cold PBS, collected in PBS, centrifuged at 845*g* for 5 min at 4°C, and lysed in Nonidet lysis buffer supplemented with protease inhibitors, as previously described. Post-lysate supernatants were collected at 20,000*g* for 15 min at 4°C. Equal volumes of the cleared lysates were incubated with 15–20 $\mu$l GFP-Trap Agarose (ChromoTek), pre-equilibrated in lysis buffer. The mixture was rotated for 2 h, at 4°C. The beads were then recovered by centrifugation (845*g*, 3 min) and washed four times with washing buffer (50 mM Tris–HCl, 1 mM EDTA, 0.1% Triton X-100). Beads were washed 3 times with PBS to remove the detergent traces and then resuspended in 50 $\mu$l of PBS for further processing by mass spectrometry.

20 $\mu$l of beads was transferred to 150 $\mu$l of 10 mM dithiothreitol in 50 mM ammonium bicarbonate solution in Lo-Bind tubes (Eppendorf) and incubated at 60°C for 1 h. Vials were mixed every 15 min to resuspend the beads during the reduction process. Reduced cysteine bonds were capped by adding 40 $\mu$l of 100 mM iodoacetamide in 50 mM ammonium bicarbonate and incubated in the dark for 30 min. Vials were brought into the light for 20 min before 30 $\mu$l of 50 mM ammonium bicarbonate containing 1 $\mu$g of Trypsin Gold (Promega). Samples were incubated overnight at 37°C and quenched with 20 $\mu$l of 1% formic acid in water (vol/vol) and transferred to LC-MS vials for analysis.

Digested samples were initially analysed on a Waters M-Class LC system connected to a SCIEX ZenoTOF mass spectrometer. The sample (20 $\mu$l) was injected onto a Waters nanoEase 0.18 × 20 mm Symmetry C18 trap column and then switched in line with a Waters nanoEase 0.075 × 150 mm HSS T3 nanocolumn. Peptides were eluted into the mass spectrometer using a gradient of 2 to 40% ACN over 51-min flowing at 0.5 $\mu$l/min. Peptides were analysed using an IDA-based analysis with the top 50 selected for fragmentation, with an exclusion period of 12 s. LC-MS/MS files were searched using PEAKS 12.5 (BSI) against a custom database containing the ATF6α_LD[S1P,S2Pmut] protein, using a fixed cysteine carbamidomethylation modification, and variable modifications of oxidised methionines and deamidated asparagines and glutamines. The PEAKS search was then run with a glycan search, which contains 1,828 basic glycans.

The extracts were reanalysed on a Waters M-Class LC system linked to a Waters TQ-XS triple quadrupole mass spectrometer with an IonKey interface. Peptides were injected onto a Waters nanoEase 0.3 × 50 mm C18 trap column before switching in line with a 0.15 × 50 mm HSS T3 iKey. Peptides were separated over a gradient of 2–55% ACN over 37 min flowing at 1 $\mu$l/min. The triple quadrupole monitored precursor ions of two glycosylated peptide variants and monitored characteristic oxonium ions relating to glycosylated peptides and the HVVEFGGENLYFQSAK (HVVE) backbone peptide.

Table 1 shows precursor (Q1) and product ion (Q3) information and collision energy (CE) for two glycopeptides and the HVVE backbone peptide. The product ion with an asterisk was used to

**Table 1. MRM transitions and optimized mass spectrometry parameters for target glycopeptides and backbone peptides.**

| Peptide | Q1 | Q3 | Charge | CE |
|---|---|---|---|---|
| HVVE backbone | 608.97 | 580.31, 743.37*, 855.40, 970.45 | 3+ | 25 |
| HEX8 peptide | 1,022.12 | 163.06, 204.08, 366.14* | 3+ | 40 |
| NeuAc2 peptide | 928.76 | 204.08, 274.09*, 366.14 | 4+ | 30 |

generate quantitative data. Other product ions were monitored for qualitative data only.

# Supplementary Information

# Acknowledgements

We thank the CIMR flow cytometry core facility team (Reiner Schulte and Gabriela Grondys-Kotarba) and the microscopy team (Matthew Gratian and Mark Bowen) for technical support; and Tianyu Wen and Stuart Haslam (Imperial College London) for assistance in glycobiology in early phases of this research. This study was supported by a Wellcome Trust Principal Research Fellowship to D Ron (Wellcome 224407/Z/21Z) and by the Spanish Ministry of Science, Innovation and Universities to A Ordoñez (RYC2022-035365-I).

## Author Contributions

G George: conceptualisation, data curation, formal analysis, validation, investigation, methodology, and writing—original draft, review, and editing.
HP Harding: investigation, methodology, and writing—review and editing.
R Kay: methodology.
D Ron: conceptualisation, resources, supervision, funding acquisition, investigation, visualisation, and writing—original draft, review, and editing.
A Ordoñez: conceptualisation, data curation, formal analysis, supervision, investigation, methodology, and writing—original draft, review, and editing.

## Conflict of Interest Statement

The authors declare no competing interests.

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
