## [Reviewer comments · Life Science Alliance]

Metabolite import by SLC33A1 is required for ATF6 activation during endoplasmic reticulum stress

Ginto George, Heather Harding, Richard Kay, David Ron, and Adriana Ordóñez

DOI: <https://doi.org/10.26508/lsa.202603679>

Corresponding author(s): Adriana Ordóñez, Universidad Católica de Murcia and Ginto George, University of Cambridge

Review Timeline:

Submission Date:	2026-02-26
Editorial Decision:	2026-03-20
Revision Received:	2026-03-26
Accepted:	2026-03-27

Scientific Editor: Tim Fessenden

Transaction Report:

Please note that the manuscript was reviewed at *Review Commons* and these reports were taken into account in the decision-making process at *Life Science Alliance*.

Review
COMMONS

Review #1

****Summary:****

In this manuscript, the authors follow up on the results from a previous CRISPR screen in CHO-K1 cells demonstrating that knockout of the ER acetyl-CoA transporter Slc33a1 suppresses ATF6 activation. The authors show in these cells that, in response to 2-DG, the Slc33a1 deletion results in constitutive activation of the UPR except for the ATF6 pathway, which appears to traffic constitutively to the Golgi but to not be cleaved there. They show using an uncleavable ATF6 that loss of Slc33a1 delays formation of an O-glycosylated form of at least this version of the protein, and they also find that single deletion of the ER acetyltransferases NAT8 and NAT8B also constitutively activates the UPR, but that activation in this case includes activation of ATF6. The mechanism by which Acetyl-CoA might impact ATF6 activation is not elucidated.

****Major Comments:****

The following conclusions are well-supported:

- That loss of Slc33a1 results in IRE1 and PERK activation but not ATF6 activation
- That ATF6 traffics at least to some degree constitutively to the Golgi when Slc33a1 is deleted, which is a counterintuitive finding given the apparent lack of ATF6 activation
- That loss of Slc33a1 can alter the level O-glycosylation and the preponderance of sialylated N-glycans on at least ATF6
- Generally speaking, I find the wording to be careful and precise

The following claims are less convincing:

- That loss of Slc33a1 results in universal suppression of ATF6 activation. The effect in response to 2-DG is unquestionably strong at least at the level of Bip-GFP reporter (although it's not clear from this paper nor the previous one from this group how much of the Bip promoter this reporter encodes- which is important because only a minimal Bip promoter is exclusively responsive to ATF6). However, the impairment of ATF6 activation in response to tunicamycin (Fig. 1C) is very modest, and no other stressors were tested (DTT and TG were used for other purposes, not to test ATF6 activation). One might actually expect this pathway, if it affects glycosylation pathways, to be particularly sensitive to a stressor like 2-DG that would have knock-on effects on glycosylation. Admittedly, it does seem to be true in the basal condition (i.e., absent an exogenous ER stress) that IRE1 and PERK are activated where ATF6 is not. At some level, it's hard to reconcile the almost complete suppression of Bip-GFP induction in Slc33a1 cells in response to 2DG with the fact that in Fig. 3, cleavage clearly seems to be occurring, albeit to a lesser extent
- That regulation of ATF6 is a broadly applicable consequence of Slc33a1 action. Unless I've missed it, all experiments are performed in CHO-K1 cells, so how broadly applicable this pathway is not clear.
- That loss of Slc33a1 "deregulated activation of the IRE1 branch of the UPR." It is clear that IRE1 is activated when Slc33a1 is deleted (that the authors show this repeatedly in different parental cell

lines provides a high degree of rigor). However, at least through the CHOP-GFP reporter, PERK is activated as well. Although 4u8C suppresses this activation, the suppression is not complete, there are no orthogonal ways of showing this (e.g., loss of KD of IRE1), and the converse experiment (examining IRE1 activation when PERK is lost or inhibited) was not done. Thus, while I agree that the data shown are consistent with PERK activation being downstream of IRE1, they are not definitive enough to, in my opinion, rule out the more parsimonious explanation for their own data and what is already published in the field that loss of Slc33a1 causes ER stress (thus in principle activating all 3 pathways of the UPR-including ATF6 transit to the Golgi) but that it also, separately, inhibits activation of ATF6 (and possibly other things? See below)-a possibility acknowledged towards the end of the Discussion.

- That "Nat8 and Slc33a1 influence ER homeostasis and ATF6 signaling through distinct mechanisms". This conclusion would require simultaneous deletion of both Nat8 and NAT8B because of possible redundancy/compensatory effects.

- If I'm understanding the authors' argument correctly, they seem to be invoking that the ATF6 activation defect underlies/is upstream of the activation of IRE1 in Slc33a1 KO cells. But if that understanding is correct, it seems fairly unlikely, as the authors' data show no evidence that ATF6 is activated in parental cells under basal conditions (Fig. 3B) and thus no reason to expect that failure to activate ATF6 by itself would result in appreciable phenotype in cells-an idea also consistent with the general lack of phenotype in ATF6-null MEF and other cells.

****Minor Comments:****

- The alteration in O-glycosylation levels of ATF6 is interesting, but it might or might not be relevant to ATF6 activation, and if it isn't, then the paper provides no mechanism for why loss of Slc33a1 has the effects on ATF6 that it does. What about other similar molecules, like ATF6B (surprising that this was not examined), SREBP1/2, a non-glycosylatable ATF6, and/or one of the other CREB3L proteins?

- Does Slc33a1 deletion cause other ER resident proteins to constitutively mislocalize to the Golgi?

- As mentioned above, does loss/knockdown of Slc33a1 activate IRE1 and PERK but not ATF6 in other cell types?

- Also as mentioned above, how do the UPR (all 3 branches) in cells lacking Slc33a1 respond to TG or DTT? This and the preceding comments are important toward making the claim that Slc33a1 is actually a regulator of ATF6. The time required to do these experiments will depend on whether creation of more stable lines is required, and whether they are worth doing depends on how broad the authors wish the scope of the paper to be.

- It's surprising that the authors didn't do comparable experiments to what is shown in Fig. 6 but deleting the acetyltransferases that modify sialic acids, which I believe are known.

- The authors mis-describe the data from Fig. 5B. EndoH and PNGaseF should collapse ATF6 to a 0N form, not a 1N form (what is labeled as 2N should be 1N, and it looks like the true 2N band is partially obscured by the strong 3N band).

****Referee cross-commenting****

While reviewer #2 and I have somewhat different opinions on the strength of the evidence, we seem fairly well-aligned on the overall significance of the work.

The conceptual advance in this paper is that, while loss of Slc33a1 seems widely disruptive to ER function—an idea that has been advanced in the literature before—it seems to have unique and discordant effects on ATF6 relative to the other UPR pathways. The paper does not offer a conclusive mechanism by which these effects are realized, and the sole focus on ATF6 makes it difficult to fully contextualize the findings, but the data are of high quality and, while the scope is somewhat narrow, the phenotype is likely to be of interest to those concerned with ER stress and UPR signaling, which also describes my own expertise.

Review #2

Summary

The authors employed a genome-wide CRISPR-Cas9 screen to search for the genes selectively involved in the activation of ER stress sensor ATF6. Deletion of Slc33a1, which encodes a transporter of acetyl-CoA into the ER lumen, compromised the ATF6 pathway (as assessed by BiP::GFP reporter), while IRE1 and PERK were activated in basal conditions, in the absence of ER stress (as assessed by XBP1s::mCherry reporter and endogenous XBP1s and CHOP::GFP reporter). Moreover, IRE1, but not ATF6, replied to ER stress.

Consistently, in Slc33a1 Δ cells upon ER stress the levels of the processed N-ATF6 α were significantly lowered compared to the parental cells, and microscopy study showed that in Slc33a1-deficient cells ATF6 is translocated to Golgi even in the absence of ER stress, but fails to reach the nucleus even after ER stress is imposed. Golgi-type sugar modification of ATF6 α is decreased in Slc33a1 Δ cells. These data show the importance of SLC33A1 for ATF6 processing and functioning through the mechanism which remains to be revealed.

Major comments.

Taken together, the reported data do support the conclusion about the role of SLC33A1 functioning in post-ER maturation of ATF6. Data and methods are presented in a reproducible way. Still, there are several issues worth attention.

1. While BiP::GFP reporter is very useful, it would be more convincing to show the level of BiP in Slc33a1 Δ cells by WB.
2. Another concern is the state of Slc33a1 Δ cells. While adaptation is a general problem of clonal cells, the cells used in this study (with XBP1 highly spliced, CHOP upregulated, and ATF6 pro-survival pathway inhibited) are probably very sick, and the selection pressure/adaptation is very strong in this cell line. I would suggest the authors to clarify this issue.
3. Authors showed that, based on CHOP::GFP reporter data, PERK was activated in the absence of ER stress and the activation was due to IRE1 signalling. But did PERK reply to the ER stress?
4. An important question is a subcellular location of SLC33A1. Huppke et al. (cited in the manuscript) showed that FLAG- and GFP-tagged SLC33A1 was colocalized with Golgi markers. While that may be due to overexpression of the protein, it deserves consideration, given that ATF6 is stuck in Golgi upon depletion of SLC33A1.
5. OPTIONAL. Regarding the role of acetylation in compromising ATF6 function: since both SLC33A1 deficiency and depletion of Nat8 have broad effects, glycosylation of ATF6 upon depletion of Nat8

should be assessed (similarly to Fig 5), to demonstrate the difference in glycosylation pattern upon the absence of SLC33A1 and Nat8 and strengthen the conclusions.

****Minor comments.****

1. Apart from the table of the cell lines, it would be useful to add to the supplementary a simple-minded scheme of the reporters used in this study (BiP::GFP, CHOP::GFP, XBP1s::mCherry) specifying the mechanism of the readout and the harbored protein and other important details (e.g., whether mRNA of XBP1s::mCherry reporter could be processed by IRE1).
2. Fig 2B and Fig 3A - the percentage of spliced XBP1 in parental cells is about 30% according to the graphs, but it looks more like 5%.
3. Fig 3B - It would probably be better to demonstrate the processing of endogenous ATF6. It could help to avoid the problems with alternative translation (even though anti-ATF6 antibodies are known to be tricky).
4. In Fig 4B - could be better to show Golgi marker separately. In Fig 4B and E the bars are missing (and cells in Fig 4B look bigger than in Fig 4E). Magnification of the insets should be further increased.
5. As the authors mention, 2-deoxy-D-glucose (2DG) is known to be the ER stress inducer, acting via prevention of N-glycosylation of proteins. Also, N-glycosylation state of ATF6 has been suggested to influence its trafficking. Thus, even if the control cells were treated in the same way, 2DG may not be the best ER-stress inducer to study ATF6 trafficking. Indeed, altered sugar modification of ATF6 α in Slc33a1 Δ cells (Fig 5) was tracked using Thapsigargin.
6. Minor comment on Fig 7 - recent data (Belyy et al., 2022) suggest IRE1 is a dimer even in the absence of ER stress.

****Referee cross-commenting****

I agree with Reviewer 1 that the authors need to clarify that authors need to clarify better how exactly BiP::GFP reporter works and whether it reflects ATF6 activation (rev 1 pointed to unclear responsiveness of the reporter to ATF6 and I asked to show the level of BiP by WB and the scheme of the mechanisms of readouts of the reporters)

I also agree with the comment on 2-DG which for some experiments may not be the best choice to activate UPR (or as Reviewer 1 pointed out shouldn't be the only one used to induce UPR). I still think that there's no contradiction in partial cleavage of ATF6 and suppression of BiP::GFP in Slc33a1 Δ cells if then (as authors show) it doesn't reach nucleus.

General assessment. The article shows the necessity of SLC33A1, a transporter of acetyl-CoA in ER lumen, for ATF6 processing and functioning. It is well-written. However, the molecular mechanism which underlies the link is yet to be discovered (and this is clearly mentioned by the authors).

The study is of interest for the basic research and of potential interest for clinical research.

My main field of expertise is UPR. While I have broad knowledge and interest in protein science in general, my experience with protein glycosylation is rather limited.

Manuscript number: RC-2025-03131

Corresponding author(s): Ginto George and Adriana Ordoñez

1. General Statements

We thank the reviewers for their careful evaluation of our work and for their constructive and insightful comments. We are pleased that both reviewers found the study to be well executed, clearly presented, and of interest to the ER stress and UPR community. We have carefully considered all comments and revised the manuscript accordingly. We believe these revisions have substantially strengthened the clarity, robustness, and conceptual impact of the study.

Below we provide a detailed, point-by-point response to the reviewers' comments and describe the revisions and new data included in the revised manuscript.

Reviewer 1 & 2 (common points)

1. Description of the BiP::GFP reporter as a readout of ATF6 α activity.

- **Comment:** Both reviewers are concerned about whether BiP::GFP is a reliable and specific reporter for ATF6 α activation.
- **Response:** In response, we have clarified in the revised manuscript the details of the BiP promoter fragment used in this reporter, explicitly detailing the presence of an ERSE-I element motif (CCAAT-N₉-CCACG), the most specifically and robustly activated by ATF6 α (new Supplemental Fig. S1). This reporter was first characterised in our recently published study (Tung *et al.*, 2024 *eLife*), where we demonstrated that BiP::GFP expression is ATF6 α dependent, as CRISPR/Cas9-mediated disruption of endogenous ATF6 α resulted in a marked reduction in BiP::GFP fluorescence compared with parental cells. Furthermore, treatment with ER stress in the presence of Ceapin-A7 (a small molecule that blocks ATF6 α activation by tethering it to the lysosome) effectively blocked activation of the ATF6 α fluorescent reporter, whereas the S1P inhibitor partially attenuated the BiP::sfGFP signal in stressed cells (Tung *et al.*, 2024 *eLife*; Supplemental S1D). We have now reproduced these findings in the present study, further confirming that the BiP::GFP reporter is highly dependent on ATF6 α activation, and we present these data in a new Supplemental Fig. S1B.

2. Correlation between BiP::GFP reporter activity and BiP expression levels.

- **Comment:** Both reviewers requested correlation of the BiP::GFP reporter activity and endogenous BiP levels.
- **Response:** To address this point, we have measured BiP mRNA levels in parental and *Slc33a1*-depleted cells under both basal conditions and ER stress conditions. These

measurements correlated well with the BiP::GFP reporter activity assessed by flow cytometry and are shown in Supplemental Fig. S3E.

3. Does ATF6 α respond to other ER stressors in *Slc33a1*-deleted cells?

- **Comment:** Both reviewers accepted our claim that ATF6 α activation is partially attenuated in *Slc33a1*-deleted cells exposed to ER stressors tunicamycin (Tm) and 2-Deoxy-D-glucose (2DG) but raised the possibility that ATF6 α signalling might respond differently to other ER stressors.
- **Response:** To address this point, we have performed new experiments assessing ATF6 α activation (BiP::GFP activity) in both *Slc33a1*-deleted and parental cells in response to additional ER stressors, including dithiothreitol (DTT) and thapsigargin (Tg). These new data, presented in a new Supplemental Fig. S3B and S3C, show that *Slc33a1*-deletion also attenuates ATF6 α signalling in cells treated with dithiothreitol (DTT) and thapsigargin (Tg).

4. Deletion of all NAT8 family members.

- **Comment:** Both reviewers suggested that deletion of all NAT8 family members was required to conclusively distinguish their role from that of SLC33A1.
- **Response:** We agree with this assessment and have now generated cells in which both *Nat8* and *Nat8b* are simultaneously deleted. These new data, included in a new Supplemental Fig. S9, strengthen the comparison with SLC33A1 deficiency and rule out potential redundancy among NAT8 family members. Notably, simultaneous inactivation of *Nat8* and *Nat8b* resulted in the same phenotype observed upon single *Nat8* deletion, namely activation of both the IRE1 and ATF6 α branches of the UPR. These findings (discussed in detail) are consistent with previous studies implicating protein acetylation in ER proteostasis but suggest that a defect in protein acetylation is unlikely to contribute to the consequences of SLC33A1 deficiency in terms of ATF6 α signalling.

5. Generalisability beyond CHO-K1 cells.

- **Comment:** Reviewer 1 raised concerns regarding validation of our findings beyond CHO-K1 cells.
- **Response:** While we acknowledge that validation in additional cell types would further strengthen the study, we now explicitly discuss the technical challenges encountered when attempting to generate clonal *Slc33a1* knockouts in aneuploid human cell lines, such as HeLa. This limitation is now clearly acknowledged in the revised version, and our conclusions are framed accordingly.

6. Relationship between basal ATF6 and IRE1 signalling.

- **Comment:** Both reviewers argued that BiP::GFP does not appear to be active under basal conditions in parental cells, and therefore a failure to activate ATF6 would not be expected to affect the conditions of the cells basally. Thereby questioning how attenuated basal ATF6

activity in the SLC33a1 deleted cells could account for the derepression observed in the IRE1 pathway.

- **Response:** The logic of the reviewer's critique is impeccable, and we thank them for the opportunity to clarify this important issue. Whilst the basal fluorescent signal arising from BiP::GFP (the ATF6 α reporter) is indeed weak, it is not null. This is evident by comparing the BiP::GFP signal in wildtype and ATF6 α -deleted cells (new Supplemental Fig. S1B) These experiments revealed a significant reduction in basal BiP::GFP fluorescence in ATF6 α Δ cells compared with parental dual-reporter cells, indicating that the BiP::GFP reporter has basal activity that is dependent on ATF6 α . These new data are consistent with previous published observations demonstrated that treatment with Ceapin, an ATF6 α -specific inhibitor, lowered BiP::GFP fluorescence in tunicamycin-treated cells to levels below those observed in untreated controls (Tung et al., eLife 2024). Together these observations indicate that ATF6 α is active basally in CHO-K1 cells. Given the established cross-pathway repression of IRE1 by ATF6 α signalling, it renders plausible our suggestion that the basal activation of the XBP1::mCherry (IRE1-reporter) observed basally in the SLC33a1 deleted cells arises from the partial interruption of ATF6 α signalling.

Reviewer 1 (additional points)

1. Effect of deleting sialic acid-modifying acetyltransferases.

- **Comment:** Reviewer 1 suggested that comparing the consequences of deleting SLC33a1 and the sialic acid- modifying acetyltransferases that operate downstream of the putative acetyl-CoA transporter could be informative.
- **Response:** In response to this valuable suggestion, we have now examined the impact of deleting *Casd1*, the gene encoding the Golgi acetyltransferase responsible for modifying sialic acids on ATF6 α activity, comparing the consequences to *Slc33a1* deletion. New Supplemental Fig. S8 reveals partial phenotypic overlap between the two deletions, suggesting that the loss of SLC33A1 exerts some of its effects on CHO cells by compromising sialic acid modification.

2. Potential effects on ATF6-like proteins (SREBP1/2, CREB3L).

- **Comment:** Reviewer 1 suggested that we evaluate the effect of SLC33A1 loss on other ATF6-like transcription factors.
- **Response:** We took this advice to heart, but our attempts to compare SREBP2 processing in wildtype and SLC33A1 knockout cells were frustrated by the low quality of the antibodies available to us.

Reviewer 2 (additional points)

1. Physiological state and clonal adaptation of Slc33a1-deleted cells.

- **Comment:** Reviewer 2 raised concerns regarding the physiological state of the *Slc33a1*-deleted cells and the potential impact of clonal adaptation or selection pressure on the consequences of genetic manipulation.
- **Response:** This is a valid concern. Deconvoluting direct from indirect effects are a challenge in any genetics-based experiment. To try and address this point, we compared the proliferation capacity of three pairs of parental CHO-K1 clones with their derivative *Slc33a1*-deletion variants using the IncuCyte assay. As shown in new Supplemental Fig. S2D, the *Slc33a1* deletion variants had no consistent fitness disadvantage revealed by this assay. Whilst cell mass accretion is only one measure of comparability between cell lines, we deem these observations to indicate that a comparison between SLC33A1 wildtype and mutant CHO-K1 cells is unlikely to be compromised by gross underlying differences in cell fitness.

2. Responsiveness of PERK signalling to ER stress.

- **Comment:** Reviewer 2 asked whether PERK signalling, which appears basally activated due to higher basal IRE1 signalling in the *Slc33a1*-deleted cells, remains responsive to ER stress.
- **Response:** To address this point, we treated cells with ER stressors and assessed PERK pathway activation. As shown in new Supplemental Fig. S4C, PERK signalling remains functional and responsive to ER stress in *Slc33a1*-depleted cells.

In addition to the points above, we have addressed several presentation and clarity issues raised by the reviewers, including figure labelling, image presentation, and schematic models. The Discussion has also been revised to more explicitly acknowledge the current limitations of the study while emphasising its central conceptual advance: namely, that loss of SLC33A1 results in a discordant UPR state in which IRE1 and PERK are activated, whereas ATF6 α trafficking and transcriptional output are selectively compromised.

The following table summarises the major changes made to the figures in the revised manuscript to facilitate tracking the modifications introduced

Figure	Figure Panels	Amendment (if any)
Fig 4	4B (modified)	Scale bar added.
Fig 5	5B (modified)	Labelling correction according to the reviewer.
Fig S1 (new)	S1A-S1B	New data detailing the BiP promoter fragment and the reliability of the BiP::GFP reporter as a readout for ATF6 α activity in cells.
Fig S2 (modified)	S2D (new)	New IncuCyte data added.
Fig S3 (modified)	S3B, S3C and S3E (new)	Panels B and C: New data from DTT and thapsigargin treatments, respectively. Panel E: New data from BiP mRNA levels under 2DG treatment in parental and Slc33a1 -deleted cells.
Fig S4 (new)	S4C (new)	Panels A and B: Previously shown as panels in Fig. S2C and S2D.

Full Revision

		Panel C: New data on the PERK response to ER stress in Slc33a1 -deleted cells.
Fig S7 (new)	S7A-S7C (new)	New sanger sequencing chromatograms displaying the targeted exonic regions of the Casd1 , Nat8 and Nat8b .
Fig S8 (new)	S8A-S8B (new)	Casd1 -deleted data added.
Fig S9 (new)	Unique panel	New data comparing Nat8/Nat8b -deleted cells with single Nat8 -deleted cells.

We thank the reviewers again for their insightful comments, which have significantly strengthened the manuscript. We believe the revised study clarifies key mechanistic points and provides a stronger conceptual advance regarding the role of SLC33A1 in UPR regulation.

Sincerely,

Adriana Ordóñez

March 20, 2026

RE: Life Science Alliance Manuscript #LSA-2026-03679

Dr. Adriana Ordóñez
Universidad Católica de Murcia (UCAM)
HiTech
Campus de los Jerónimos 135, E-30107,
Murcia, Murcia E-30107
Spain [ES]

Dear Dr. Ordóñez,

Thank you for submitting your revised manuscript entitled "Metabolite import via SLC33A1 enables ATF6 activation by endoplasmic reticulum stress". Your manuscript was returned to original reviewers from Review Comments and their comments are below. As you will see, both reviewers are satisfied overall and include suggestions to further improve this work. In particular Reviewer 1 notes a potential error in Fig S6 and recommends citing a recent preprint. While we concur with Reviewer 2 on the value of reporting protein levels of endogenous Bip, including these data is left to your discretion. In view of the reviewer support and in accordance with our prior commitment, we would be happy to publish your paper in Life Science Alliance pending these optional changes and final revisions necessary to meet our formatting guidelines.

MANUSCRIPT ORGANIZATION AND FORMATTING:

To avoid unnecessary delays in the acceptance and publication of your paper, please read the following information carefully. Full guidelines are available on our Instructions for Authors page, <https://www.life-science-alliance.org/authors>

- Please upload your main manuscript text as an editable doc file.
- Please add a Running Title and a Summary Blurb/Alternate Abstract in our system.
- While the current title is acceptable, we encourage you to consider this slight change: "Metabolite import by SLC33A1 is required for ATF6 activation during endoplasmic reticulum stress"
- Please add a Category for your manuscript in our system.
- Please add the X and Bluesky handles of your host institute/organization, as well as your own, and/or one of the authors, in our system.
- Please mark the Secondary Corresponding Author in our system as well.
- It is recommended to exclude figures from the manuscript text and upload them separately.
- Please add Authors Contributions to our system as well.
- Please rename "Disclosure and competing interest's statement" to "Conflict of Interests."
- Please add your main and supplementary figure legends to the main manuscript text after the references section.
- Please add callouts for Figures S5A-B; S6A-C and S8A-B to your main manuscript text.
- Please add molecular weight to the blots in Figures 2E, 3A, and S4A.
- We suggest uploading Figure 7 as a Graphical Abstract. If you agree, kindly remove this figure file and upload with the proper designation.
- Thank you for including a Key Reagents table in your Materials and Methods section. It is advised to upload this table as a Supplementary File, and refer to it at the end of the Materials and Methods section.

We welcome submissions of potential cover images for the issue of LSA in which your work would appear. If you have high quality images associated with this work, please feel free to email these, with a caption, to the journal office.

LSA encourages authors to provide a 30-60 second video where the study is briefly explained. We will use these videos on social media to promote the published paper and the presenting author (for examples, see <https://docs.google.com/document/d/1-UWCfbE4pGcDdcgzcmiuJI2XMBJnxKYeqRvLLrLSo8s/edit?usp=sharing>). Corresponding or first-authors are welcome to submit the video. Please submit only one video per manuscript. The video can be emailed to contact@life-science-alliance.org

FINAL FILES:

The following items are required for acceptance.

The license to publish form must be signed before your manuscript can be sent to production. A link to the license to publish form will be available to the corresponding author only. Please take a moment to check your funder requirements.

Thank you for your attention to these final processing requirements. Please revise and format the manuscript and upload materials as soon as you are able.

Thank you for this interesting contribution to the literature. We look forward to publishing your paper in Life Science Alliance.

Sincerely,

Reviewer #1 (Comments to the Authors (Required)):

In this revised manuscript, the authors address a number of the concerns raised by the other reviewer and me, citing technical limitations for others. The revision is fairly responsive, although a couple of the issues left open would not have seemed so difficult to resolve: many other cell lines are tractable to CRISPR knockout besides CHO-K1 (it's unclear why the authors cite a need for human cell lines in the rebuttal), and examining endogenous activation of UPR pathways in Slc33a1-knockout cells, rather than having to create reporter lines, would seem a reasonable approach. Likewise, whether the response affects transit and cleavage of other ATF6-like molecules could have been addressed by transgene expression of an epitope-tagged version of one of the other factors (akin to what was done for ATF6a in Fig. 4D). And testing whether Slc33a1 deletion causes other ER resident proteins to relocate to the Golgi would seem relatively trivial. Nonetheless, the manuscript is written in a sufficiently conservative way that its limitations are not glossed over, so I believe the paper should be accepted with only a few minor revisions:

1. Figure 6SB, the peak area values for the backbone peptide are duplicated from those of Hex8 for the parental (I presume it is the peptide values that are wrong, because with those values the quantification in C doesn't make sense)
2. It is odd in Fig. S3B that DTT appears to barely activate the XBP1s-mCherry reporter at all, which is inconsistent with the fairly extensive body of literature that DTT is a potent IRE1 activator and inducer of Xbp1 splicing. Do the authors have an explanation for this?
3. Since this revised manuscript was submitted, a paper was uploaded to BioRxiv from the Wiseman group (Kutseikin et al)

showing activation of IRE1/XBP1 signaling upon putative Slc33a1 pharmacological inhibition with the IXA4 agent. It is worth citing and briefly remarking on this study.

Reviewer #2 (Comments to the Authors (Required)):

Summary

The authors employed a genome-wide CRISPR-Cas9 screen to search for the genes selectively involved in the activation of ER stress sensor ATF6 in CHO cells. Deletion of Slc33a1, which encodes a transporter of acetyl-CoA into the ER lumen, compromised the ATF6 pathway (as assessed by BiP::GFP reporter and BiP mRNA levels), while IRE1 and PERK were activated in basal conditions, in the absence of ER stress (as assessed by XBP1s::mCherry reporter and endogenous XBP1s and CHOP::GFP reporter). Moreover, IRE1 and PERK, but not ATF6, replied to ER stress.

Consistently, in Slc33a1 Δ cells upon ER stress the levels of the processed N-ATF6 α were significantly lowered compared to the parental cells, and microscopy study showed that in Slc33a1-deficient cells ATF6 is translocated to Golgi even in the absence of ER stress but failed to reach the nucleus even after ER stress was imposed. Golgi-type sugar modification of ATF6 α was decreased in Slc33a1 Δ cells.

These data demonstrate the importance of SLC33A1 for ATF6 processing and functioning. Although the exact mechanism remains to be revealed, acetyltransferases of NAT8 family have been shown not to be responsible for the phenotype of Slc33a1 Δ cells while deletion of CASD1, Golgi-resident sialic acid O-acetyltransferase, leads to milder but similar consequences, suggesting the involvement of CASD1.

Comments.

I support the publication of this manuscript. The only minor comment remaining at this point: I would still recommend showing the data on BiP protein level changes for overall clarity.

Role of SLC33A1 on ATF6 α #LSA-2026-03679-R1**Detailed point-by-point response to issues raised in review.**

The original reviewer's comments are in BLACK and our response in PURPLE.

Reviewer #1:

In this revised manuscript, the authors address a number of the concerns raised by the other reviewer and me, citing technical limitations for others. The revision is fairly responsive, although a couple of the issues left open would not have seemed so difficult to resolve: many other cell lines are tractable to CRISPR knockout besides CHO-K1 (it's unclear why the authors cite a need for human cell lines in the rebuttal), and examining endogenous activation of UPR pathways in Slc33a1-knockout cells, rather than having to create reporter lines, would seem a reasonable approach. Likewise, whether the response affects transit and cleavage of other ATF6-like molecules could have been addressed by transgene expression of an epitope-tagged version of one of the other factors (akin to what was done for ATF6a in Fig. 4D). And testing whether Slc33a1 deletion causes other ER resident proteins to relocate to the Golgi would seem relatively trivial. Nonetheless, the manuscript is written in a sufficiently conservative way that its limitations are not glossed over.

We thank the reviewer for the careful evaluation and constructive suggestions. We appreciate the points raised regarding additional not human cell lines, evaluation of UPR activation without endogenous reporters or relocation of proteins in Slc33a1 depleted cell lines. These are interesting and valuable directions. However, we feel that addressing them would require substantial additional experimental work that is beyond the scope of the current study and the revisions undertaken here. We are pleased that the reviewer finds the revised manuscript sufficiently clear and appropriately cautious, and we appreciate the recommendation for acceptance.

Minor revisions:

1. Figure 6SB, the peak area values for the backbone peptide are duplicated from those of Hex8 for the parental (I presume it is the peptide values that are wrong, because with those values the quantification in C doesn't make sense).

We thank the reviewer for identifying this mistake in Fig. S6B, as by error, the peak area values for the backbone peptide were duplicated for parental. We apologise for this inaccuracy; it has now been corrected in the revised manuscript and the values in Fig. S6B are consistent with the quantification data.

2. It is odd in Fig. S3B that DTT appears to barely activate the XBP1s-mCherry reporter at all, which is inconsistent with the fairly extensive body of literature that DTT is a potent IRE1 activator and inducer of Xbp1 splicing. Do the authors have an explanation for this?

This is a valid point, however while IRE1 activation mediated by DTT is readily detected by changes in phosphorylation and XBP1 splicing, reporter assays such as XBP1s::mCherry and BiP::GFP rely on new protein synthesis and therefore have a narrower dynamic range. This is likely due to the concurrent inhibition of protein synthesis under DTT treatment, which can limit reporter signal despite pathway activation. Moreover, given the short duration of the 3 h DTT treatment, substantial new protein synthesis would not be expected.

3. Since this revised manuscript was submitted, a paper was uploaded to BioRxiv from the Wiseman group (Kutseikin et al) showing activation of IRE1/XBP1 signaling upon putative Slc33a1 pharmacological inhibition with the IXA4 agent. It is worth citing and briefly remarking on this study.

We thank the reviewer for drawing our attention to the Wiseman's work, which is now cited in the discussion section of the revised manuscript.

Reviewer #2:

The authors employed a genome-wide CRISPR-Cas9 screen to search for the genes selectively involved in the activation of ER stress sensor ATF6 in CHO cells. Deletion of Slc33a1, which encodes a transporter of acetyl-CoA into the ER lumen, compromised the ATF6 pathway (as assessed by BiP::GFP reporter and BiP mRNA levels), while IRE1 and PERK were activated in basal conditions, in the absence of ER stress (as assessed by XBP1s::mCherry reporter and endogenous XBP1s and CHOP::GFP reporter). Moreover, IRE1 and PERK, but not ATF6, replied to ER stress.

Consistently, in Slc33a1 Δ cells upon ER stress the levels of the processed N-ATF6 α were significantly lowered compared to the parental cells, and microscopy study showed that in Slc33a1-deficient cells ATF6 is translocated to Golgi even in the absence of ER stress but failed to reach the nucleus even after ER stress was imposed. Golgi-type sugar modification of ATF6 α was decreased in Slc33a1 Δ cells.

These data demonstrate the importance of SLC33A1 for ATF6 processing and functioning. Although the exact mechanism remains to be revealed, acetyltransferases of NAT8 family have been shown not to be responsible for the phenotype of Slc33a1 Δ cells while deletion of CASD1, Golgi-resident sialic acid O-acetyltransferase, leads to milder but similar consequences, suggesting the involvement of CASD1.

Comments.

I support the publication of this manuscript. The only minor comment remaining at this point: I would still recommend showing the data on BiP protein level changes for overall clarity.

We thank the reviewer for their positive assessment of our manuscript and for the helpful suggestion. Regarding BiP protein levels, we note that in our experience, quantification by immunoblot can be less reliable and more variable than measuring BiP mRNA levels. For this reason, and to maintain consistency with our current dataset, we have focused on BiP transcript measurements, which we consider a more robust readout in this context. We therefore have not included additional BiP protein-level analyses, as these would require much further experimental work.

March 27, 2026

RE: Life Science Alliance Manuscript #LSA-2026-03679R

Dr. Adriana Ordóñez
Universidad Católica de Murcia
HiTech
Campus de los Jerónimos 135, E-30107,
Murcia, Murcia E-30107
Spain

Dear Dr. Ordóñez,

Thank you for submitting your Research Article entitled "Metabolite import by SLC33A1 is required for ATF6 activation during endoplasmic reticulum stress". It is a pleasure to let you know that your manuscript is now accepted for publication in Life Science Alliance. Congratulations on this interesting work.

Your article will publish open access upon publication under a CC-BY license.

DISTRIBUTION OF MATERIALS:

Again, congratulations on a very nice paper. I hope you found the review process to be constructive and are pleased with how the manuscript was handled editorially. We look forward to future exciting submissions from your lab.

Sincerely,
